# Snail-inspired robotic swarms: a hybrid connector drives collective adaptation in unstructured outdoor environments

Da Zhao [1,2], Haobo Luo[1,2], Yuxiao Tu[1,2], Chongxi Meng[1,2] & Tin Lun Lam [1,2] ✉

Terrestrial self-reconfigurable robot swarms offer adaptable solutions for various tasks. However, most existing swarms are limited to controlled indoor settings, and often compromise stability due to their freeform connections. To address these issues, we present a snail robotic swarm system inspired by land snails, tailored for unstructured environments. Our system also employs a two-mode connection mechanism, drawing from the adhesive capabilities of land snails. The free mode, mirroring a snail's natural locomotion, leverages magnet-embedded tracks for freeform mobility, thereby enhancing adaptability and efficiency. The strong mode, analogous to a snail's response to disturbance, employs a vacuum sucker with directional polymer stalks for robust adhesion. By assigning specific functions to each mode, our system achieves a balance between mobility and secure connections. Outdoor experiments demonstrate the capabilities of individual robots and the exceptional synergy within the swarm. This research advances the real-world applications of terrestrial robotic swarms in unstructured environments.

Robot swarms draw inspiration from natural swarm systems observed in various species, such as fish[1–3], insects[4–6], and birds[7]. In nature, these organisms exhibit collective behaviors that enable them to accomplish tasks that are far beyond the capabilities of any individual member. By working together in a coordinated manner, swarms can display emergent properties and abilities that individual members do not possess. This remarkable feature of swarm intelligence allows these groups to achieve greater efficiency, robustness, and adaptability, ultimately enhancing their chances of survival and success in their respective environments[8]. Building on inspiration from these natural swarm systems, recent years have seen a surge of research in aerial[9], terrestrial[10], and aquatic[1] robot swarms. These studies not only contribute to a deeper understanding of animal behavior but also explore the potential of utilizing swarm robotics to perform various tasks. In contrast to their aerial and aquatic counterparts, terrestrial robot swarms have faced limitations in their applicability to outdoor environments. The majority of their designs have been centered around operation on flat, indoor surfaces, which has inadvertently constrained their potential for broader applications. To unlock the full capabilities

of terrestrial robot swarms, it is crucial to consider and address the challenges associated with navigating diverse and unstructured outdoor terrains.

The transition from indoor to outdoor environments presents different challenges for aerial, aquatic, and terrestrial robot swarms. For aerial and aquatic swarms, the hardware requirements remain relatively consistent, as the mediums they operate in are generally uniform and unchanging, with obstacle avoidance being the primary concern. However, terrestrial robot swarms face unique challenges due to their constant interaction with the ground. While numerous indoor terrestrial robot swarms have been developed for operation on flat surfaces[11–15], they are often ill-suited for unstructured environments with steps, ditches, and varying surface materials. In recent years, the only specially designed multi-legged robot swarm[10] for outdoor environments has demonstrated the ability to traverse steps, gaps, and other outdoor terrains. However, this type of robot can only connect end-to-end, making it essentially a 2D robot swarm. The connection strength between robots is relatively weak, and they can only form single chain-like structures. As a result, their obstacle-crossing

[1]School of Science and Engineering, The Chinese University of Hong Kong, Shenzhen, Shenzhen, China. [2]Shenzhen Institute of Artificial Intelligence and Robotics for Society, Shenzhen, China. ✉e-mail: tllam@cuhk.edu.cn

capabilities are quite limited, whether dealing with steps or gap-like terrains. 3D modular self-reconfigurable robot swarms are a fascinating technology that offers many advantages over traditional robots[16–18]. However, most modern modular self-reconfigurable systems either lack individual mobility or are primarily restricted to operating in controlled indoor environments[19–30]. In the context of large-scale deployment in outdoor environments, reconfigurable terrestrial robot swarms exhibit significant potential for operation in unstructured settings. These adaptable robots, when functioning as individual units, demonstrate the capability to explore and maneuver in diverse outdoor scenarios. The emphasis on the single robot's field mobility ensures the overall system's flexibility and agility. Furthermore, the incorporation of a robust connection mechanism becomes pivotal, ensuring that when the robot swarm assembles into a cohesive unit, it attains heightened robustness. This multifaceted approach highlights the synergy between individual mobility and robust interconnectivity, crucial for the successful deployment and operation of reconfigurable terrestrial robot swarms in dynamic outdoor environments (Fig. 1).

When developing 3D self-reconfigurable terrestrial robot swarms, the choice of interconnection method among robots holds utmost importance. Freeform, like chain, lattice, truss, and hybrid, stands as a fundamental structural category for modular robot swarms[17,18]. Compared to fixed-position connectors requiring precise dock-to-dock alignment, such as retractable mechanical hooks[20,27], permanent magnets[10,22,25], electromagnets[31], and self-soldering alloys[32], freeform connectors generally offer a much larger acceptance area[33]. This broader acceptance proves crucial for large-scale robot swarm deployment, addressing challenges like low sensor precision, manufacturing inaccuracies, and structural deformations[34]. In recent years, research has surged around 2D[11,35–37] and 3D[34,38–40] freeform robot swarms with high connection success rates. FreeBOT, with its fully

spherical shell, is an exceptional freeform robotic system; however, it faces challenges such as a weak single-point connection and limited vertical friction. To address these issues, Zhao and Lam proposed SnailBot[39], a sliding sphere-type robot swarm. Despite improvements, neither FreeBOT nor SnailBot offers sufficient connection strength for tasks like angle-based locomotion or manipulation, which require fixed configurations and considerable shear friction between robots. A potential solution involves a connection mechanism with greater strength, akin to FireAnt3D[34]. However, FireAnt3D's connector has limitations, such as a limited number of cycles and low efficiency. FreeSN[40], the first heterogeneous freeform modular robot, comprises *Node* and *Strut* modules. With a parallel-type structure, FreeSN effectively carries blocks and overcomes obstacles. However, its single module has limited mobility in outdoor environments.

To develop a freeform self-reconfigurable terrestrial robot swarm suitable for outdoor environments, two primary challenges must be addressed. The initial challenge involves designing a robot with a freeform connector that can effectively operate outdoors. In our pursuit of solutions, we turn to nature for inspiration, leveraging the collective behaviors observed in swarms. Nature's examples, such as ants forming bridges to cross gaps or gullies[41] and simple components yield high-level behaviors in biological organisms[42,43], showcase the emergence of remarkable capabilities through collective actions. However, these nature-inspired swarms often lack the mobility required for navigating unstructured environments[11,34]. Recognizing this limitation, we propose exploring land snails. Land snails are gastropod mollusks that possess a unique anatomy[44,45], allowing them to climb walls, overcome barriers, and navigate uneven surfaces. We develop a 3D freeform self-reconfigurable snail robot swarm for field applications, which draws inspiration from the unique anatomical structure of snails. The morphological evolution from a snail to a snail robot is depicted in Fig. 2a. The snail robot's design incorporates the

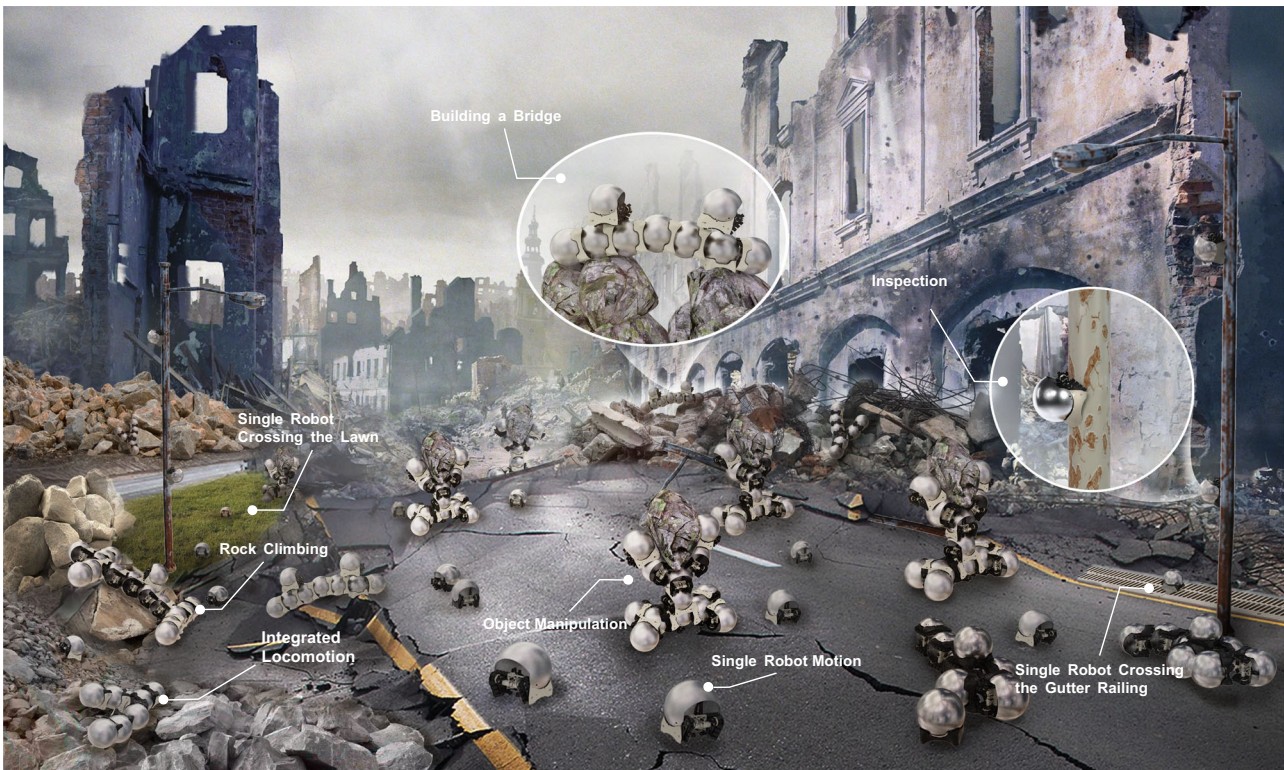

**Fig. 1 | Overview of the snail robot swarm system.** A depiction of the mission profile: A single snail robot can traverse most outdoor terrains and even climb metal poles to carry out monitoring tasks. When working together, multiple robots can effectively navigate various types of landscapes, such as step-like, trench-like, and other challenging terrains. Additionally, these robot swarms can assemble themselves into robotic arms for object manipulation (see Supplementary Movie 1).

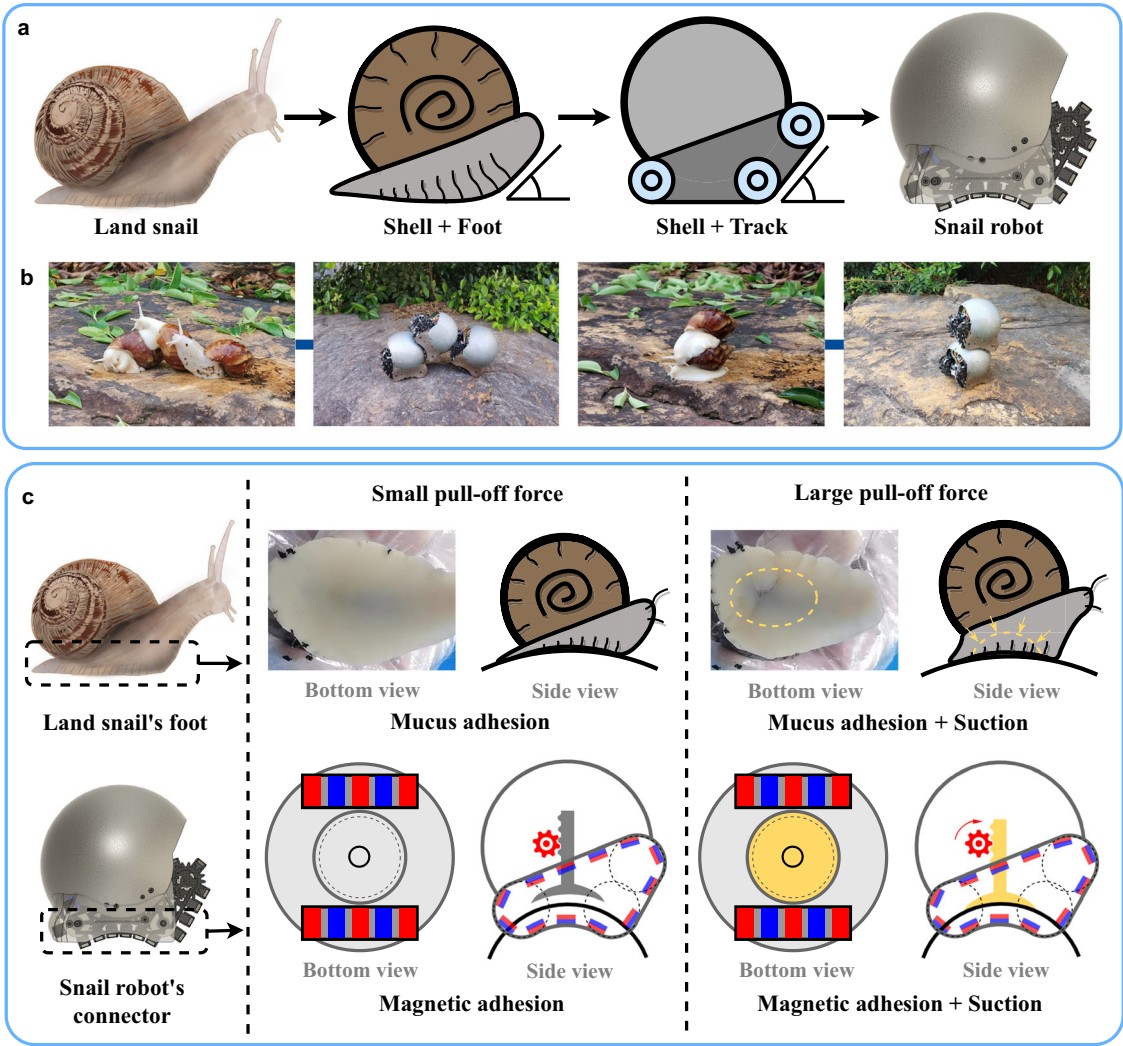

**Fig. 2 | Overview of the inspiration. a** Morphological evolution from a snail to a snail robot. **b** Snails outdoors may climb onto another snail's shell for various reasons. Inspired by this, snail robots can also interconnect with each other. **c** Comparative illustration of snail foot adhesion and snail robot connection mechanism under varying external forces (see Supplementary Movie 2).

two primary components of a snail's body, including the spherical shell and the foot. Furthermore, in nature, snails congregate and even attach to each other for various reasons, such as mating[46], moisture retention, and temperature regulation. The snail robot features the ability to connect to another robot's ferromagnetic spherical shell using its connection mechanism, which resembles snails' attachments in nature (Fig. 2b). This expandable capacity enables the formation of larger, more adaptable robotic systems capable of handling a broader array of tasks.

The second challenge involves designing an efficient and stable connector for the snail robot swarm. Traditionally, connectors for self-reconfigurable robots are typically classified as mechanical couplers or magnetic couplers based on the acting forces[47]. Here, we classify them based on the level of freedom provided, dividing them into discrete connection (dock-to-dock connection, such as refs. 20,23), free connection (connect at any position, but the connection point cannot change once connected, such as ref. 34), and free transition (connect anywhere and can seamlessly adjust the connection point location, such as refs. 38,39). We strive to develop the third type of connector, which boasts the utmost level of freedom. However, this type of connector frequently faces limitations in connection strength. Drawing from nature's blueprint, the snail robot employs a dual-mode connection mechanism akin to that of a real snail, as illustrated in Fig. 2c.

Real snails use mucus adhesion to adhere to substrates[48,49], enhancing their suction force when they encounter an external pulling force, as demonstrated in experimental studies on snail's adhesion characteristics[50]. Similarly, the snail robot uses magnetic adhesion to connect to other robots' spherical shells and to transition between robots. When a substantial force is exerted on the robot's shell, such as when other snail robots connect to it, the robot extends its vacuum sucker to generate a suction force, to ensure a secure attachment to another robot's spherical shell. In doing so, we can create a hybrid connector that delivers the highest degree of freedom while maintaining the capability to establish robust connections as required.

In this work, we aim to design and develop a snail-inspired robotic swarm system with dual-mode connection mechanisms that can efficiently execute a variety of tasks in unstructured environments. Initially, we analyze the morphology of the snail and its dual-mode connection mechanism to cope with varying external forces. Based on this knowledge, we subsequently establish the fundamental design principles for the snail robot, including its basic form and connection mechanisms. Following this, we present a detailed overview of the snail robot's design. We then allocate tasks under different modes and outline the principles for mode switching. This is further expanded with an analysis of the properties of both the free and strong modes. Thereafter, we conduct extensive testing of a single robot's terrain

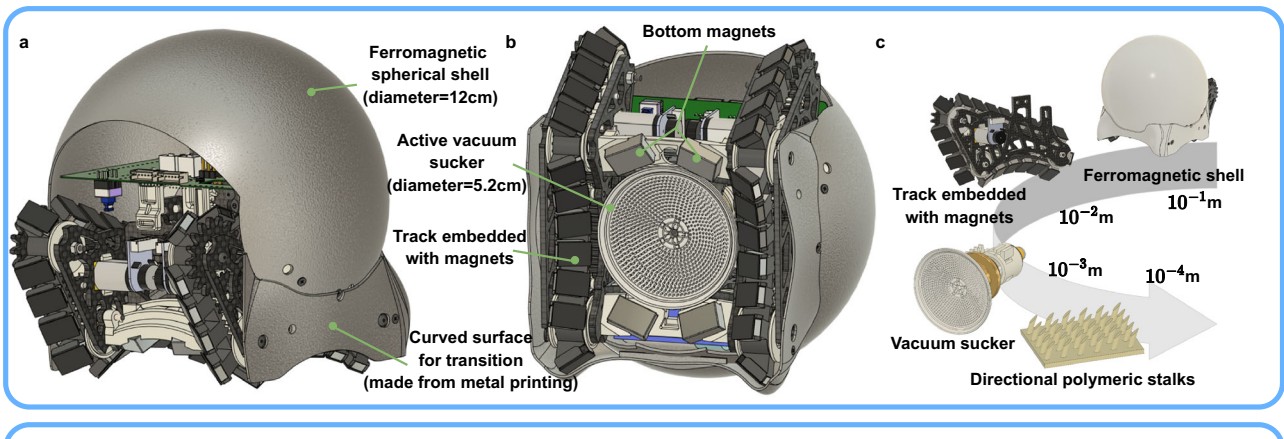

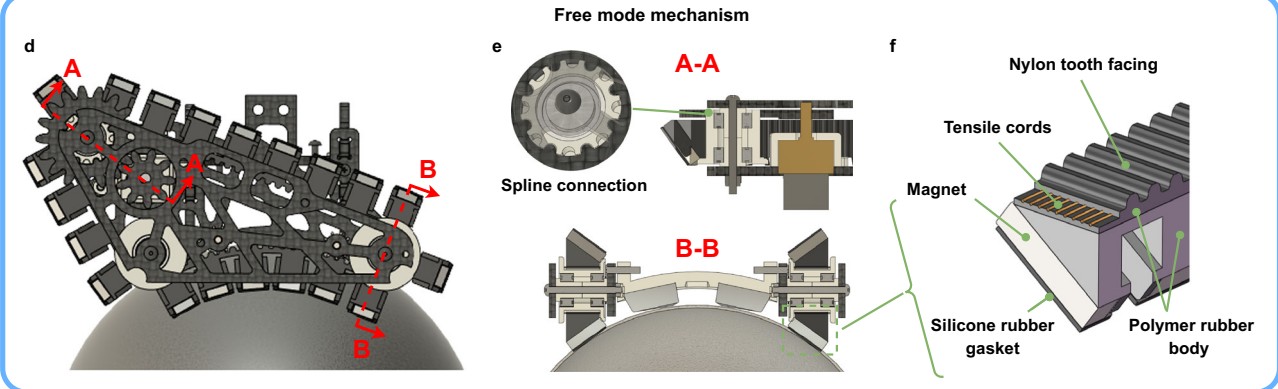

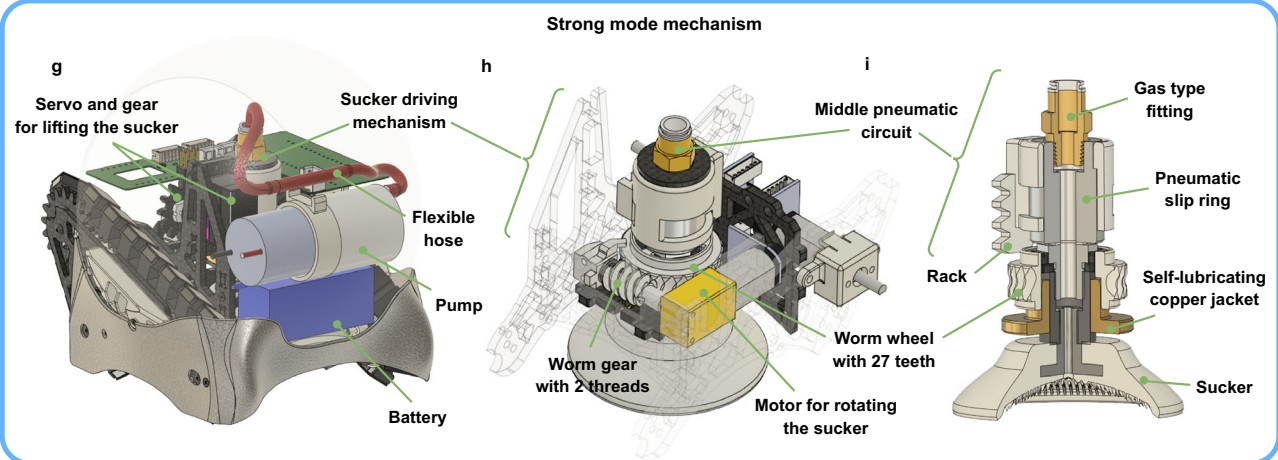

**Fig. 3 | Mechanism overview of a single snail robot. a** Isometric view. **b** Bottom view. **c** Elements of a robot over a range of length scales. **d** Side view of a full track. **e** The cross-section of the tracks. **f** Main elements of a track. **g** The mechanism inside the spherical shell. **h** Sucker driving mechanism. **i** Middle pneumatic circuit.

traversing capabilities. Ultimately, we showcase the ability of the snail robot swarm to execute tasks, including traversing various terrains and performing manipulative tasks.

## Results

### Mechanism overview

A snail robot primarily consists of a ferromagnetic spherical shell and a dual-mode connection mechanism, as illustrated in Fig. 3a. The snail robot utilizes magnetic tracks as its primary propulsion mechanism to enable locomotion, eliminating the need for additional actions such as lifting the rocker as in the initial version. This approach reduces the complexity of both the structure and control. Furthermore, the magnetic tracks effectively combine the six-wheel suspension and bottom magnets of the previous version, freeing up valuable space at the bottom to install other connectors. The magnets on the tracks enable

the robot to adhere to another robot's spherical shell. To enhance the connection strength, a retractable vacuum suction cup with directional polymeric stalks is employed at the bottom, as shown in Fig. 3b. Robot components across various length scales are displayed in Fig. 3c, ranging from the spherical shell with a 120 mm diameter at a scale of $10^{-1}$ m, to the directional polymeric stalks (DPS) with a 600-micron diameter at a scale of $10^{-4}$ m.

A snail robot features a freeform magnetic connection and a fixed suction connection, as demonstrated in Fig. 3d–i. In freeform magnetic connection, the robot primarily relies on two tracks with embedded magnets, as illustrated in Fig. 3d–f. Many prior magnetic climbing robots employed chain tracks with magnets on each chain link; however, this structure is often heavy and unsuitable for small swarm robots. In this design, lightweight soft polymer rubber and synchronous belts compose the tracks, with magnets embedded in the

polymer rubber. The rubber tracks' elasticity renders them more compatible with a spherical surface. The cross-section of the tracks, displayed in Fig. 3e, f, consists of nylon teeth, cords, and part of the polymer rubber body, forming a standard synchronous belt. The silicone rubber gasket increases the friction between the tracks and the shell. The configuration of the magnets' pole faces is strategically designed to create an angle with the rotational axis of the track wheel, thereby facilitating tangency with the shell.

The mechanism employed in fixed adhesion connection is depicted in Fig. 3g–i. When the snail robot requires a strong connection, it extends the suction cup to make contact with the spherical shell below, using a gear rack mechanism driven by a servo motor. A negative-pressure air pump is installed on the robot's main body and is connected to the middle rotary pneumatic circuit through a flexible hose. When the robot is in strong connection mode and rotates, the suction cup is always connected to the lower spherical shell. Therefore, a single-channel pneumatic slip ring is installed in the rotating pneumatic circuit to ensure that the suction cup can rotate 360° without losing the negative pressure provided by the air pump. Since the negative pressure inside the suction cup can be rapidly neutralized after the air pump is shut off, the lifting mechanism does not need to provide a large lifting force. A worm gear mechanism drives the suction cup's rotation relative to the robot body (Fig. 3h). Owing to the worm gear mechanism's self-locking property, the central motor need not supply additional holding force once the robot enters the strong connection mode and the torsion resistance of the suction cup can be utilized even if the rotary drive motor is not operating. The worm gear's transmission ratio was meticulously calculated to equalize the maximum angular velocity supplied by the differential tracks and the suction cup rotating motor during yaw rotation. We did explore the possibility of extending the suction cup further to enhance adhesion on flat surfaces. This would provide increased stability and reduce the risk of tipping. However, after careful consideration, we decided against this modification. Given that our snail robot group primarily operates outdoors in varied and complex environments, achieving consistent and tight adhesion to overly complex surfaces proved challenging.

## Dual-mode task allocation and switching principle

Typically, tasks performed by a terrestrial robot swarm can be broadly categorized into six classes: assembly/disassembly, self-reconfiguration, flow, manipulation, locomotion, and support, as depicted in Fig. 4a. The first three types demand individual mobility and the ability of robots to move freely on their peers. The connection strength required for these tasks is generally not high and it suffices to support a single robot unit. However, the latter three types usually involve multiple robots linked together or forming a cantilevered structure, scenarios where a mere free connector often struggles to provide stability. To address these diverse needs, we have engineered two distinct operating modes for snail robots: free mode and strong mode. Our goal is to strategically assign these modes to robots based on the nature of their assigned tasks, thereby optimizing their performance and efficiency in various scenarios.

In free mode, a snail robot uses its differential tracks with embedded magnets to facilitate a free magnetic connection. The robot is capable of executing three primary actions in this mode: yaw, sliding, and transitioning between modules. The first three tasks, assembly/disassembly, self-reconfiguration, and flow, can be performed in this mode. The magnetic connection provides the robot with a level of agility and smooth movement across the surface of other robots, enhancing the swarm's overall adaptability. Therefore, the free mode empowers our snail robot to handle tasks that require a high degree of adaptability and flexibility, offering a promising approach to swarm robotics in an environment of uncertainty.

In strong mode, a snail robot utilizes its retractable vacuum suction cup to form a high-strength suction connection. This enhancement mechanism not only fortifies the module's vertical anti-torque ability but also increases yaw drive torque, supporting the latter three tasks— manipulation, locomotion, and support. By entering the strong mode, the robots can significantly boost their connection strength, allowing for more demanding tasks. Hence, strong mode expands the range of tasks that the snail robot swarms can handle, enhancing their ability to form robust and complex structures when necessary.

Figure 4b depicts the equivalent joints created by two robots in both modes. In free mode, the snail robot leverages its free magnetic connector, thereby attaining three degrees of freedom. Such configuration endows a snail robot connected to its counterpart with the ability to perform yaw rotation and sliding actions, facilitating its maneuverability. In this context, the equivalent joint comprises a universal joint and a rotary joint (i.e., a spherical joint). In comparison to conventional spherical joints, the equivalent joint between two robots represents a spherical joint with a nonholonomic constraint due to the velocity constraint imposed by the differential tracks. When a robot transitions to a specific position on another robot's spherical shell and switches to the strong mode, it can only perform yaw rotation. From the standpoint of equivalent joints, the strong mode locks the universal joint while allowing the rotary joint to remain active. Figure 4c presents the complete control architecture of the system, incorporating detailed control logic for mode switching.

Figure 4d demonstrates the principles governing the switch between the two modes. In free mode, a robot has three degrees of freedom, providing more flexible movement; however, this mode's connection strength is comparatively weak, generally supporting just one robot. When the end of a snail robot must connect to multiple peers, it switches to strong mode and activates the strong connection mechanism. In this mode, the robot is limited to yaw movements. The primary objective of mode switching is to sacrifice degrees of freedom to attain increased connection strength. Consequently, based on the mode-switching logic, as displayed in Fig. 4d, six robots (excluding the base) form a double-arm robotic arm. The four snail robots proximal to the base must be in strong connection mode since their ends are linked to other robots. The two robots at the end (robot 1 and robot 2) can operate in either free mode or strong connection mode. when the end of the arm that robot 1 is on needs to be joined by a new robot to create a longer arm, robot 1 switches from free mode to strong mode. Subsequently, robot 2 in free mode can traverse other robots and relocate to robot 1's top via several transitional actions. At this point, only robot 2 and robot 3 among the six robots can function in free mode.

## Free mode connection and motion

When in free mode, the snail robot slides between other peers. The smoothness and reliability of a single robot's reconfiguration action can make the whole system more efficient. To accomplish self-assembly, self-disassembly, and self-reconfiguration tasks, a snail robot must perform fundamental reconfiguration actions, including connection, separation, and transition (subdivided into adjacent and non-adjacent transitions). The connection action enables robot swarms to merge from dispersed individual movements on the ground into a larger robot, whereas the detachment action reverses this process. Each robot's transition action alters the connection relationships within the entire robot system, facilitating the self-reconfiguration task of transforming from one configuration to another. A sequence of continuous transition actions by multiple robots creates a flow motion. Various constraints exist on the dimensional parameters and magnetic force of a snail robot to successfully and robustly execute these four basic reconfiguration actions. For instance, the area that the tracks cover on the spherical shell must not be excessively large, as it would affect the number of connectable robots at the top of a robot. The connecting magnetic force between robots should be neither too

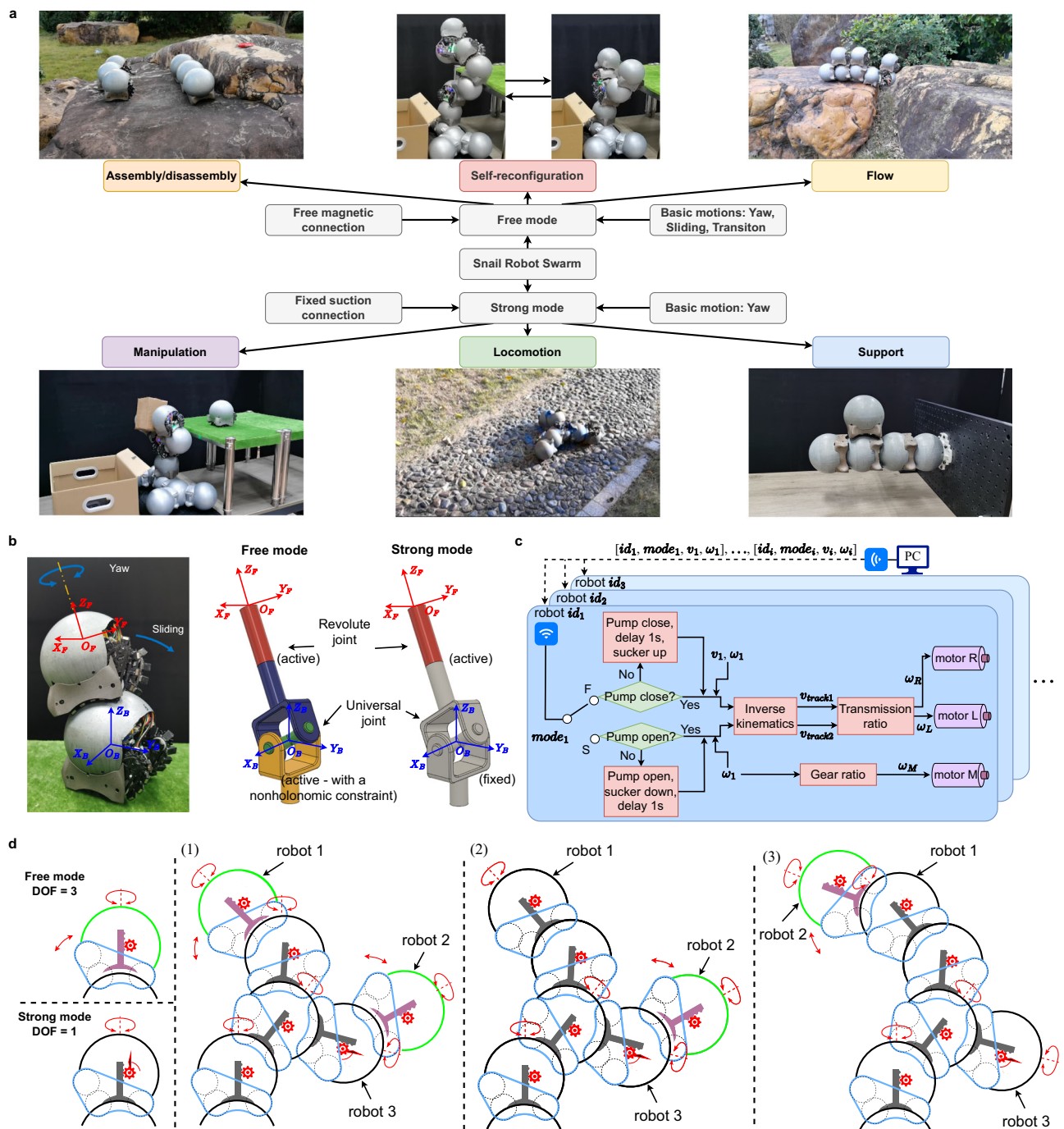

**Fig. 4 | Task allocation in two modes and their switching principle. a** The snail robot swarm possesses six capabilities. In free mode, a single module employs two differential tracks with embedded magnets to perform the three fundamental actions of yaw, sliding, and transition, thereby enabling the module to accomplish assembly/disassembly, self-reconfiguration, and flow functions. In strong mode, a single module uses fixed adhesion to increase connection force and yaw drive torque, which facilitates manipulation, locomotion, and support. **b** Equivalent joints between two robots in the two modes. **c** Overall system control architecture. **d** Two modes switching principle.

strong, to prevent failure to detachment, nor too weak, as it could fail to support a module's weight, leading to disconnection. The detailed parameter optimization process is described in Supplementary Note 1. Using an optimization-based approach, we determined the robot's size.

To successfully execute the four fundamental reconfiguration actions in free mode, two critical factors must be considered: (1) The driving capability must be adequate at every moment of movement; (2) Passive connection failure between modules should not occur

(referring to tipping over or slipping). Active tipping over or slipping is not considered a failure, such as when a robot module actively detaches from another robot module. We can conclude that the successful execution of the four basic reconfiguration actions depends on the robot experiencing resistance less than its driving force and avoiding passive connection failure. Consequently, we define three risk ratios: (1) Resistance Risk Ratio (RERR), representing the proportion of the robot's experienced resistance to its driving force. A resistance risk ratio of 0 indicates the robot's tracks are idling without resistance,

while a ratio of 1 signifies resistance equal to the maximum driving force, rendering the robot nearly immobile. (2) Tipping Over Risk Ratio (TORR), denoting the likelihood of the robot tipping over in its current state; a value approaching 1 implies a higher probability of tipping over. (3) Slipping Risk Ratio (SPRR), expressing the risk of the robot slipping in its present state; a value closer to 1 indicates a greater likelihood of slipping. Supplementary Note 2 provides a detailed derivation process for the three risk ratios.

The variation curves for the three risk ratios when the robot performs the four basic reconfiguration actions are illustrated in Fig. 5. The horizontal axes of the four graphs depict the movement distance of the robot's tracks relative to its contact surface. When the robot moves on flat ground, this value corresponds to the advancement distance of the robot's spherical shell. Conversely, when the robot moves on the spherical shell, this value represents the length of the spherical shell arc traversed by the robot. For example, in the case of an adjacent transition action, as demonstrated in Fig. 5c, the robot's tracks cover a total distance of 95.81 mm. We employ a black dashed line to indicate the distance the robot advances in each state, clearly showing that these distances are not uniformly distributed. The maximum value for the vertical axes in the four graphs is 1, and a red dashed line is used to represent the upper bound of the risk ratio. In the connection action, the initial distance between the centers of the two robots is 150 mm. The robot advances 36.61 mm to establish contact with the spherical shell of the robot in front. The maximum values for the three risk rates occur in the state depicted in Fig. 5a(4). The TORR reaches 0.64 in this state, suggesting that the connector is nearing a tipping point but remains within the safe range. For the detachment action, the moving module progresses 130.65 mm from being directly above a module's spherical shell to fully detaching from it. The maximum values of RERR and TORR are observed in the state shown in Fig. 5b(2). In this state, if the magnetic force between the modules decreases, RERR may also decrease, but TORR will increase. Consequently, selecting the appropriate magnetic force for the magnets necessitates reducing both risk rates to safe levels. For the first transition action (Fig. 5c), the robot relocates from one spherical shell to an adjacent one, with the tracks covering a total distance of 95.81 mm. The riskiest situation throughout the process occurs at state (2), where excessive magnetic force between modules results in a high RERR, hindering the robot's forward movement and causing a reconfiguration action failure. In state (2), the robot is on the verge of tipping over. As this tipping over is actively induced by the track driving force, we do not classify it as a failure mode, and the TORR remains relatively low. For the non-adjacent transition action (Fig. 5d), the robot's tracks travel a total of 135.15 mm. When the robot advances 4.48 mm from the initial state, the foremost track of the moving robot barely reaches the spherical shell of the robot in front. At this juncture, to break free from the magnetic force between the robot and the one below, the robot's tracks need to generate a greater driving force, resulting in a substantial RERR value. The robot module's driving power might be inadequate, leading to it becoming stuck in this position. The TORR and SPRR values at state (4) closely resemble those of TORR and SPRR values in Fig. 5a(4).

## Strong mode connection and motion

Free mode facilitates free movement of the snail robot, but its connection strength is usually smaller compared to self-reconfigurable robot swarms with fixed attachment points, which can make it difficult for such freeform robots to perform some tasks, such as manipulation or locomotion. Therefore, drawing inspiration from terrestrial snails that fortify their adhesion under external forces, it's critical for our robots to have a similar capability. Accordingly, we've developed a strong mode that can be triggered when attaching another robot atop the shell, emulating the robust connection seen in snail behavior. The strong connection mode features two key design elements: Firstly,

when in this mode, the robot's connection force must be reinforced, providing increased resistance against external forces. Secondly, as the end of the robot is often connected to other robots, sufficient yaw direction driving force is required in this mode. The design of the robot's connector strength enhancement mechanism will take these two elements into consideration.

There is space between the two tracks of the snail robot where a connector reinforcement mechanism can be placed. Before designing this mechanism, it is necessary to analyze the main forces that the middle mechanism will be subjected to, so that we can strengthen it accordingly. From ref. 31 and ref. 40, we can know that the external forces that a single modular robot needs to bear can be decomposed into four parts: normal force, shear force, torsion $z$ and bending force (the representation of the coordinate axes follows Fig. 4b. The points of application for the normal force and tangential force are both in the middle of the robot connector. Torsion $z$ is applied along the robot's $z$ axis. The point of application for the bending force is at the center of the robot's spherical shell. The first three types of forces applied to modular robots can directly cause similar effects on the middle connector strength enhancement mechanism. For example, applying torque to the entire robot around the $z$ axis will only result in a smaller torque to the central connector around the $z$ axis. However, when it comes to the bending force, things are a little different. The bending force exerted on the center of the robot's spherical shell can be converted into the normal force and shear force applied at the midpoint of the robot's central connector. The conversion process can be found in Supplementary Note 3. Therefore, the robot's connector strength enhancement mechanism between the two tracks is primarily subjected to three kinds of forces: tangential force $F_s$, pull force $F_n$ along the $z$ axis, and torque $\tau_z$ along the $z$ axis. The robust connecting mechanism must be able to effectively handle these three forces.

Since the connection mechanism is a regular-shaped iron spherical shell, there are two main schemes to enhance the connection strength: strong magnet and suction cup, which are commonly used in climbing robots[51–55]. Although our choice of a suction cup solution is rooted in biomimicry, it's important to note that, given our application scenarios, this solution presents numerous advantages over the magnetic scheme. For magnetic adsorption, the robot can use liftable permanent magnets or switchable electromagnets. However, the magnet solution generally adopts a relatively large volume to have a large adsorption force. At the same time, to resist large shear force in the horizontal direction and torque in the vertical direction, it is usually necessary to pad some materials that enhance the coefficient of friction under the magnet. This will weaken the magnetic force instead because the magnetic force decays quickly when the magnet is about to move away from the attracted metal[56]. In addition, if a large permanent magnet is used, in order to control the magnitude of the magnetic force, a lifting mechanism with a large pulling force is required. If the electromagnet solution is used, a lot of energy is also required to magnetize the electromagnet. However, the immediate implementation of a conventional suction cup within a snail robot could lead to technical complications. As referenced earlier, the central connection enhancement mechanism is designed to manage three distinct forces: shear force, normal force, and vertical torque. Considering that both the shell surface and typical suction cups possess a smooth texture, the suction cup would encounter substantial difficulties in counteracting both shear force and vertical torque. Therefore, we should consider optimizing the design of the standard suction cups to better cater to our specific requirements. In recent years, there has been a lot of research on the problem of robots or humans climbing on smooth surfaces[57–59]. Some of them employ a synthetic dry adhesion named directional polymeric stalks (DPS) that can be

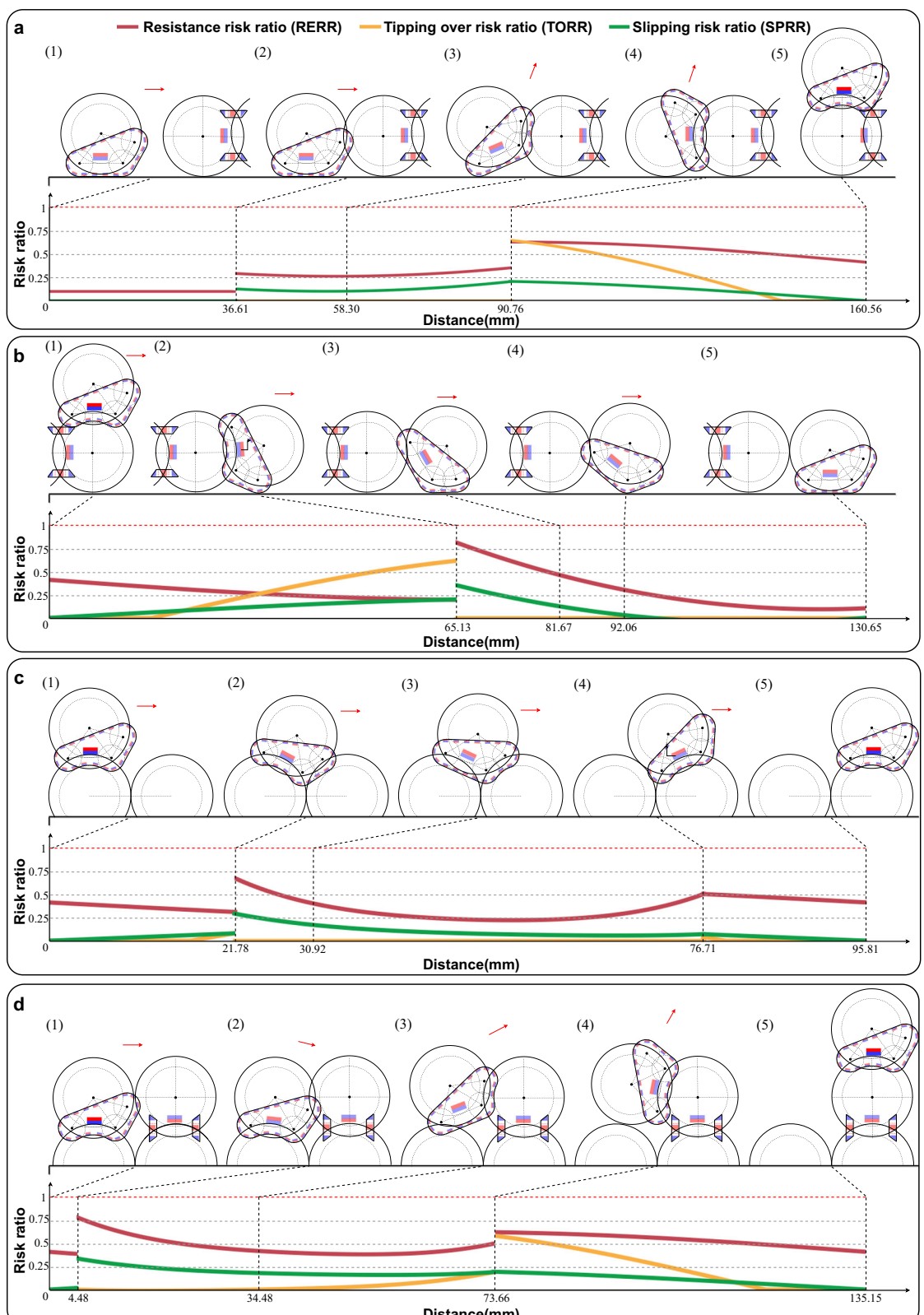

**Fig. 5 | Resistance risk ratio (RERR), tipping over risk ratio (TORR), and slipping risk ratio (SPRR) during basic actions in free mode. a** RERR, TORR, and SPRR during robot's connection action. **b** RERR, TORR, and SPRR during the robot's separation action. **c** RERR, TORR, and SPRR during the robot's adjacent transition action. **d** RERR, TORR, and SPRR during the robot's non-adjacent transition action.

adopted in our design. Gecko feet are known for their remarkable ability to stick to surfaces, even smooth or wet surfaces, without leaving a residue. The key to their adhesive properties lies in the thousands of tiny hairs or setae that cover the bottom of their feet.

These setae are split into even smaller branches called spatulae, which are only a few nanometers wide and can adhere to surfaces through weak intermolecular forces. Directional polymeric stalks mimic this structure by using synthetic materials to create an array of

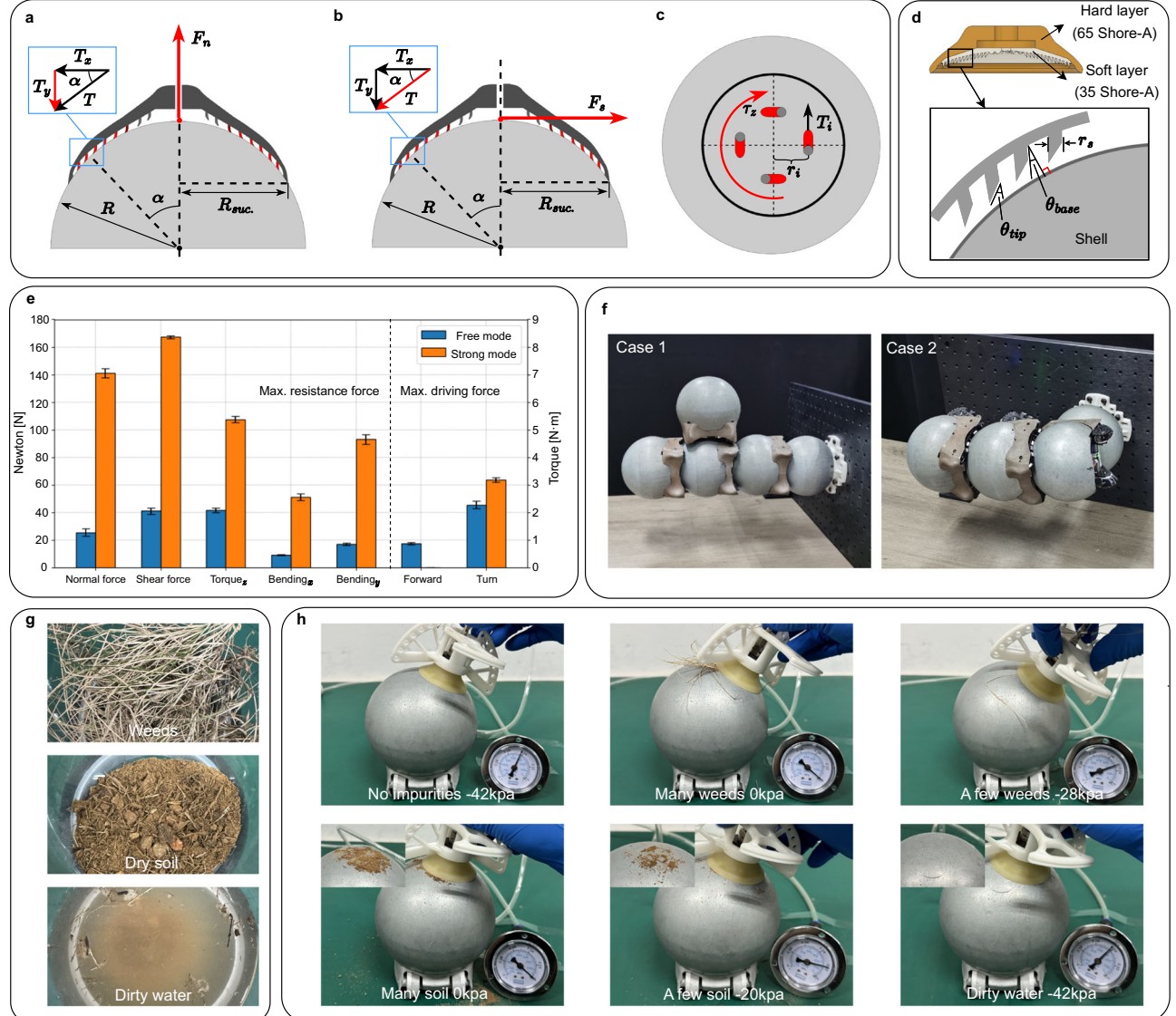

**Fig. 6 | Porperties of snail robot in strong mode. a** Analysis of DPS's reactive force against the normal force. **b** Analysis of DPS's reactive force against the shear force. **c** Analysis of DPS's reactive force against the torque along the z axis. **d** Hardness parameters of the multi-layer sucker and size parameters of the DPS. **e** Comparison of connection strength and driving capability in free mode and strong mode.

(Except for Torque and Turn, the units for other data are in N). **f** The maximum number of connectable modules for the two cantilever structures. **g** Three typical outdoor contaminants: weeds, dry soil or dust, and dirty water. **h** Impacts of different level of cleanliness on the performance of sucker.

microfibers that are angled in a specific direction. When the adhesion is pressed onto a surface, the fibers conform to the surface and create a strong bond. However, when the adhesion is peeled away in the opposite direction, the angled stalks allow it to easily release without leaving any residue. The suction cup with DPS can improve the vertical force, shear force, and torsion force to a certain extent, especially the latter two, as shown in Fig. 6a–c. As shown in Fig. 6a, when the sucker is subjected to a normal force $F_n$, the DPSs around the sucker tend to be pulled toward the middle of the sucker and thus are in a state of the load. In addition to the suction of the suckers themselves, the force component $T_y$ along the y-axis of each DPS also provides adhesion to the suckers.

The difference is when the sucker is subjected to a tangential force, as shown in Fig. 6b. The DPS away from the direction of the force $F_s$ will be in a fully stressed state. The force $T$ along the tangential direction of the shell of each stalk is used to resist the external shear force. The tangential resistance increase of the whole sucker is the sum of $T$ of all the stalks. Thanks to DPS, the suction cup can withstand

much more torque. As shown in Fig. 6c, when the sucker is subjected to a torque in the clockwise direction, the tangential force $T_i$ in the opposite direction will be received by the stalks on the sucker. The $T_i$ of each stalk and the distance $R_i$ create a moment of resistance. A comparison of the maximum normal force, shear force, and torque resistance of the suction cup with and without DPS has been shown in Supplementary Fig. 16. The detailed hardness parameters of the sucker and size parameters of DPS finally used are shown in Fig. 6d. Similarly to ref. 57 and ref. 60, we employed a multi-layer hardness strategy in our sucker design. The benefit of this approach lies in the fact that a softer contact surface is more tacky, while a harder base ensures the sucker maintains its shape despite external forces.

The comparison of connection strength and driving capability in free mode and strong mode is presented in Fig. 6e. Evidently, the strong mode, while compromising on forward movement, significantly enhances the robot's resistance to external forces and increases its output torque compared to free mode. The maximum number of connectable robots for the two cantilever structures is shown in Fig. 6f.

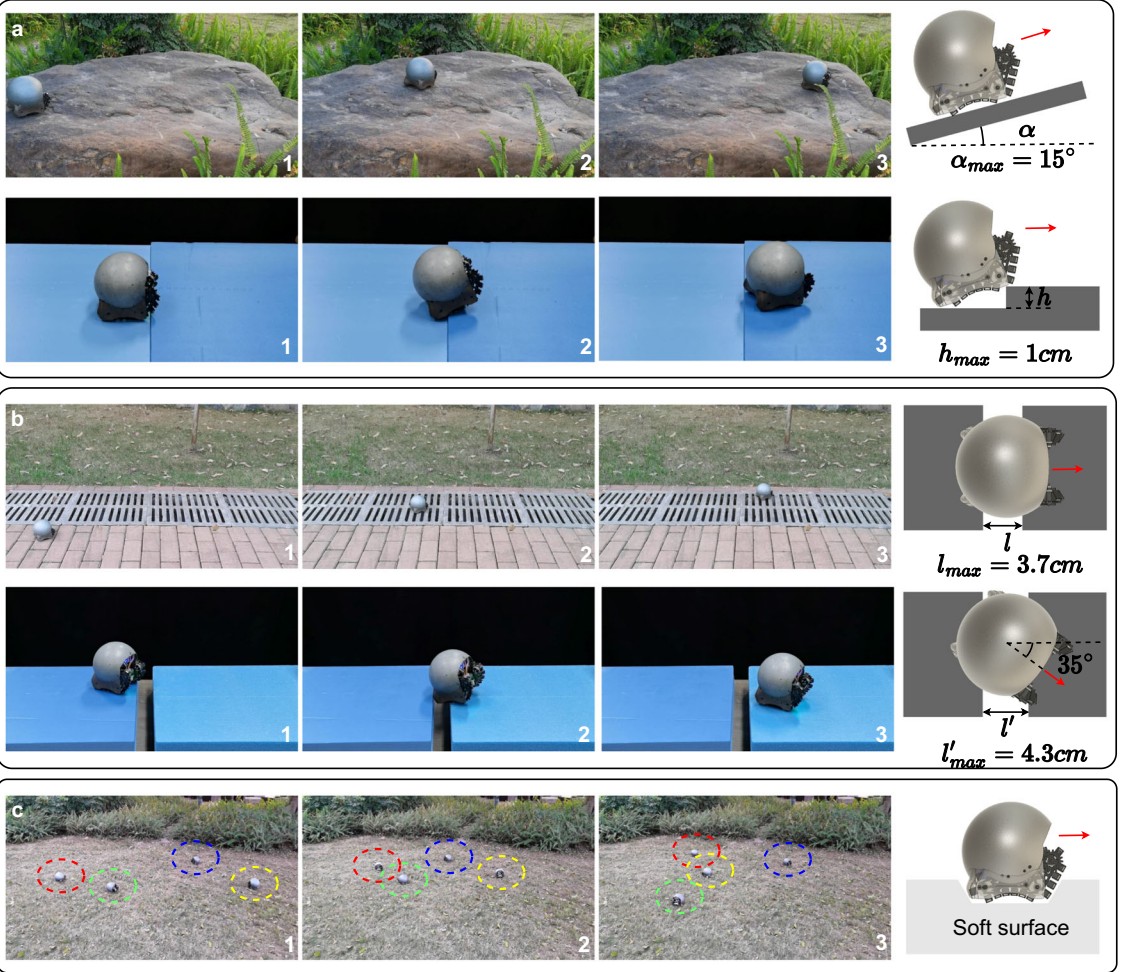

**Fig. 7 | Individual robot moves in the field. a** Snail robot moves on uneven surfaces: the robot can move on uneven outdoor rock terrain; the maximum climbing angle for the robot's forward movement is -15 degrees; the maximum step height a single robot can traverse is about 1 cm. **b** Snail robot passes gaps: the robots can pass through gutter railings outdoors; when moving directly towards a gap, a single robot can pass through gaps with a maximum width of 3.7 cm; when approaching a gap at a 35-degree angle, a single robot can pass through gaps with a maximum width of 4.3 cm. **c** Snail robot moves on deformable terrains, such as the lawn (see Supplementary movie 3).

The investigation into the impact of surface and environmental cleanliness on the performance of the strong connection mechanism is imperative. Our experiments (Fig. 6g, h) revealed that the presence of significant contaminants such as weeds or dust between the suction cup and the shell surface substantially reduced the vacuum level, leading to a loss of functionality. However, under more realistic conditions with minimal contaminants, the suction cups demonstrated resilience, maintaining approximately half of the optimal performance. Notably, the introduction of dirty water had negligible effects, indicating a degree of robustness in wet conditions. These findings underscore the importance of considering real-world environmental factors in evaluating the functionality of the vacuum suction cups, with the current design proving reasonably robust for typical operational conditions. Nevertheless, in case of working in extreme conditions, we also contemplate potential enhancements. For instance, a future design iteration could incorporate a reverse-jet functionality in the suction cup. This feature would clear impurities on the spherical shell before engaging in a strong connection, ensuring optimal performance even in extreme conditions.

### Individual robot's locomotion ability

The performance of a single snail robot is tested in both indoor and outdoor environments. The specifications of a snail robot are shown in Supplementary Table 2. For a single robot, due to the strong obstacle-crossing ability of its caterpillar tracks, it can move in outdoor environments such as lawns or concrete pavement. This is of great benefit for self-assembling into large robots in the field. We use a centralized control strategy to remote control multiple robots. The overall system control architecture is shown in Fig. 4c.

Similar to multi-legged robot swarms[10], we test the single snail robot unit in various complex environments, as illustrated in Fig. 7. Unlike some robot swarms that can only move on flat surfaces, the snail robot is capable of traversing uneven terrains. As depicted in Fig. 7a, the robot can maneuver across uneven stone surfaces. The maximum climbing angle for the robot's forward movement is -15 degrees, and the maximum step height a single unit can traverse is about 1 cm. A Snail robot can also navigate gaps, as demonstrated in Fig. 7b. In outdoor environments, the robots can pass through gutter railings. When moving directly towards a gap, a single robot can cross gaps with a maximum width of 3.7 cm, while a multi-legged robot can only traverse gaps with a maximum width of 2.5 cm. When approaching a gap at a 35-degree angle, a single snail robot can pass through gaps with a maximum width of 4.3 cm. Furthermore, snail robots can operate on deformable terrains, such as lawns, as displayed in Fig. 7c.

However, owing to size constraints, a solitary robot encounters significant obstacles when traversing certain terrain features such as

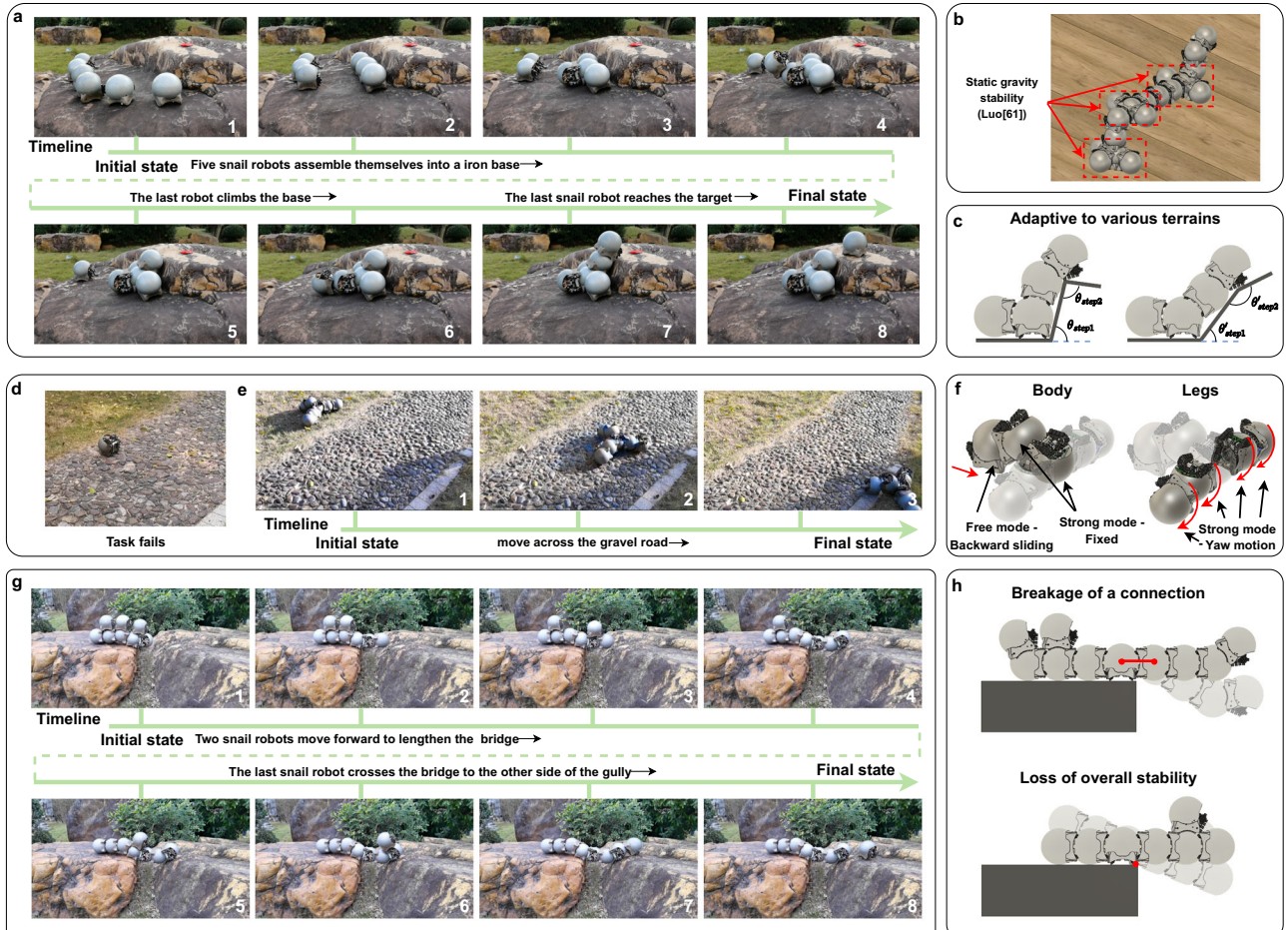

**Fig. 8 | Snail robot swarms traverse various kinds of terrains. a** Flow diagram of a snail robot swarm collaborates to climb the rock (see Supplementary Movie 4). **b** The supporting robots on both sides contribute to the overall static gravity stability. **c** The freeform connectivity feature of the snail robot enables it to adapt to steps or slopes with varying incline angles. **d** A single robot fails to traverse the cobblestone road. **e** Flow diagram of multiple snail robots collaborating to traverse the cobblestone road (see Supplementary Movie 5). **f** Each robot's mode configuration and movement status. **g** Flow diagram of eight snail robots collaborate to build a bridge to cross the gully (see Supplementary Movie 6). **h** Two types of failure.

elevated steps, wide gullies, or roads strewn with large fragments. Multiple snail robot swarms can traverse terrain that is inaccessible to an individual robot by utilizing collective flow motion or by forming a larger, integrated robot. We present four main outdoor experiments according to the order of mode selection, as shown in Supplementary Fig. 11. The connection strength and motion ability of the individual robot, as well as the gravity stability of the overall robot system, are all considered in these experiments. The mode switching logic follows the statement in Section Dual-mode task allocation and switching principle.

### Collaborate to climb a rock step

This experiment demonstrated the ability of snail robot groups to collaborate to overcome obstacles in the wild. Six modules were scattered on the ground initially. Ahead was a stone step that was 1.5 times the height of the robot, and the robots had to climb it to reach the target point at the top. Firstly, the three snail robots moved towards the edge of the step and aligned themselves horizontally along the edge of the stone step to ensure the overall structure's lateral stability (Luo provides detailed proof of static gravitational stability for this type of robot swarm in ref. 61). Next, a fourth one was connected to the middle robot of the first three units to ensure the longitudinal stability of the overall structure, as shown in the third picture of Fig. 8a. The fifth snail robot climbed onto the base formed by the first four robots and leaned against the stone steps. Finally, the sixth snail robot

climbed the iron base formed by the first five peers and achieved a separation action at the top to break free from the magnetic force and reach the top of the step. The freeform connectivity feature of the snail robot enables it to adapt to steps or slopes with varying incline angles, offering greater flexibility in navigating diverse terrains, as shown in Fig. 8c.

Multi-legged robot swarms can climb steps with a maximum height of 2.5 cm[10]. In contrast, snail robots excel remarkably in this particular task. With enough units, they could theoretically ascend steps of any height, as long as the robots at the bottom can bear the weight of their companions above.

### Traverse the cobblestone road

This experiment demonstrates the locomotion ability of a large three-legged robot made up of seven small robots. In this experiment, we initially attempted to have a single snail robot traverse a cobblestone road. However, due to the presence of some protruding cobblestones, the overall surface was rough and uneven. As a result, the individual snail robot ultimately failed to traverse this challenging terrain, as depicted in Fig. 8d.

We then employed multiple robots to form a larger robot capable of traversing this terrain. The configuration of a three-legged robot is shown in Fig. 8f. The body section primarily consists of three modules, with the top two modules in strong mode and the bottom module in free mode. The bottom module can drive the tracks to propel the

entire robot forward. When we need a larger structure, we can usually extend the length or volume of the mechanism by connecting more robots. In this way, we can get various kinds of body structures. Each leg section contains two robots, both in strong mode, to achieve greater output torque. Both modules of a leg rotate, providing more opportunities for contact with the ground and increased output torque. In this large robot, a total of five robots provide forward power. It is worth noting that the maximum torque the connectors can withstand in strong mode is limited. Therefore, the maximum output torque of a single leg is capped at this maximum resistive torque value.

As the roughness of the terrain increases, the stability requirements for the connections between modules become higher. This is because locomotion on rough terrain may demand higher rotational torque. Without suction cups to enhance stability, modules may experience instability when facing terrains like cobblestones, where the connections between modules may not withstand greater driving torque and could break. In general, on rough terrain, modular self-reconfigurable robot swarm favors configurations where the torque is biased towards the geometric center. For instance, the tripod configuration demonstrated in our demo showcases effective movement on cobblestones. This configuration allows the front two arms to exert greater torque to overcome ground friction. The detailed process of the larger robot navigating the cobblestone terrain is illustrated in Fig. 8e.

### Collaborate to build a bridge to cross the gully

The experiment demonstrated the ability of multiple snail robots to cross a trench-type obstacle. Firstly, these robots must collaborate to construct a bridge that will provide support for the robot passing through the gully. When robots build bridges, cantilever structures are formed. If the cantilever is too long, the connection may break. Therefore, there is a limit to the width of the ditch that can be crossed. In this experiment, we selected a trench formed by two boulders with a width of ~30 cm, as shown in Supplementary Fig. 12. In the previous section, we mentioned that up to 3 robots can be connected simultaneously to form a cantilever structure, with an additional robot above the middle robot. When the width of the ditch exceeds the length of the cantilever formed by three robots, adding a fourth robot to the end of the cantilever will cause the connection to break, as shown in the upper part of Fig. 8h[62]. The cantilever structure formed by snail robots is about 32 cm long, which allows the end of the cantilever to be supported by boulders on the other side of the ditch. The lower part of Fig. 8h illustrates another possible form of failure: the instability of the overall structure. Because the snail robots are untethered in outdoor environments, when the number of cantilever robots exceeds the number of robots on the side of the base, the overall structure becomes unstable. In our experiment, we used exactly four modules to balance the three-module cantilever. The robot crosses a bridge from one boulder with a relatively flat surface to another boulder with a very uneven surface. Due to the strong obstacle-crossing ability of the individual robot, the snail robot can walk on the surface of the boulder and reach the target point.

For multi-legged robot swarms, their connectors can only support one robot in a cantilevered state, limiting the maximum width they can traverse to less than the length of two robots; otherwise, the connection would break. In their experiments, the maximum indoor gap traversal demonstrated was 10 cm. In contrast, the snail robot swarm, with a comparable size to individual multi-legged robots, can traverse gaps up to 30 cm wide, owing to the adoption of a connection mode switching strategy.

### Robotic manipulation

The set of experiments demonstrates the ability of multiple snail robots to form a robotic arm to manipulate objects. In strong mode, the snail robots can be connected one by one to form a robotic arm. The arm may be single-chain or multi-chain and can be used to control other objects or even other individual robot units. In the wild, multiple snail robots can form a robotic arm and base to move other peers, as shown in Fig. 9a. It is difficult to move down the stone wall with a very large vertical height. We can use a single-chain robotic arm composed of snail robots to move the robot above the stone wall. The height of the arm needs to be higher than the stone wall so that the free module can climb to the top of the arm and be transferred to the ground by the arm. As a complement to the outdoor scene, we also show two indoor experiments. Figure 9b shows the transformation of the snail robot swarm from a single-arm to a dual-arm manipulator. Initially, the single-arm manipulator, composed of seven snail robots (excluding the base), places the terminal snail robot onto the work surface. The terminal robot, operating in free mode, then detaches from the larger manipulator arm. Subsequently, the single-arm manipulator self-reconfigures into a dual-arm structure. Following this, as shown in Fig. 9c, we also show the indoor process of picking up and placing objects with two arms composed of snail robots. At first, the two-arm manipulator leans against the block and in an open gripper position. A free snail robot on the table then pushes the block into the center of the claw. Then, the claws close, so that the two robots in free mode come together and move the block to the side of the carton. Finally, the two robots at the end in free mode move outward, that is, the jaws open, and the block falls into the interior of the carton.

## Discussion

In this study, we have presented a 3D self-reconfiguring freeform terrestrial snail robot swarm system, specifically designed to operate in unstructured environments. The development of the snail robot swarm addresses the limitations of existing terrestrial robot swarms, which are predominantly confined to indoor environments.

The introduction of the hybrid connection system, inspired by the snails' unique locomotion and adhesive capabilities, features both medium-stability magnetic connection and high-stability suction connection. This design represents a significant improvement over conventional free connection mechanisms. By leveraging two distinct modes, free and strong, the snail robot can achieve greater flexibility, adaptability, and scalability in a wide range of outdoor environments. The bionic design approach for the hybrid connection system demonstrates the substantial value of biomimicry in robotics. Not only does it provide a model for enhancing the stability and flexibility of freeform terrestrial robot swarms, but it also suggests a promising avenue for future research in developing connection mechanisms and strategies inspired by other elements of nature.

The comprehensive outdoor experiments conducted in this study serve as strong evidence of the snail robot's capabilities as a collaborative swarm, both in terms of individual unit actions and multi-unit applications. The experiments included various typical outdoor terrains such as step-like structures, rugged terrains, and uneven surfaces. These findings have important implications for the future development of terrestrial robot swarms, with potential applications in diverse real-world scenarios such as search and rescue, environmental monitoring, and infrastructure maintenance, among others.

In determining the optimal number of modules and their interconnection for specific tasks, we choose the configuration of the swarm based on the final reward of many candidate configurations controlled by reinforcement learning in the simulation environment. For example, the configuration in Fig. 8e is selected after evaluating a large number of candidate configurations in a simulated flatland environment. The rationale behind selecting this specific configuration is that the performance score gap between the configurations in the simulation environment can roughly evaluate the actual performance score gap of the real robots. From the empirical analysis of physical experiments, this configuration does have multiple advantages. For example, the two legs on both sides of this configuration can rotate at

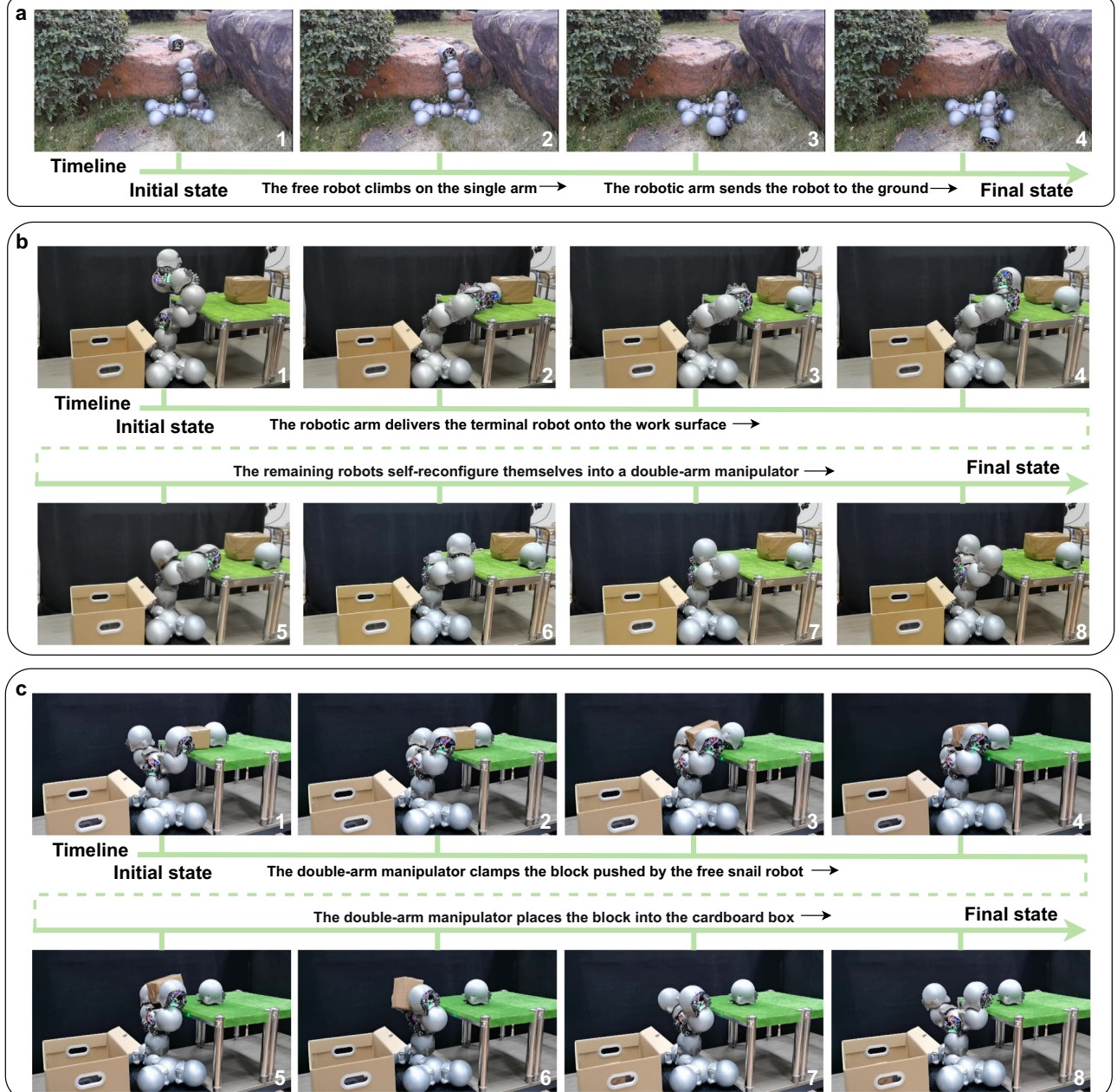

**Fig. 9 | Snail robot swarms form different kinds of robotic arms to perform manipulation tasks. a** Flow diagram of an outdoor single-arm experiment. **b** A single-arm manipulator made of snail robots can reconfigure itself into a dual-arm robot. **c** Flow diagram of an indoor dual-arm pick-and-place experiment (see Supplementary Movie 7).

differential speeds to avoid obstacles, while ensuring that the entire large robot walks stably. Methods for configuration selection and optimization similar to those described above have historically included genetic algorithms[63,64], library addressing[65], and gradient descent[66]. Recent studies leverage neural networks, such as RoboGrammar[67] and Transform2Act[68]. Challenges persist due to time-intensive evaluations and discrete optimization parameters, areas our research group aims to address in future work.

Future research will also focus on enhancing the robustness of the connection mechanisms, especially with respect to their resistance to external forces. In addition, efforts will be made to expand the types of integrated robots that the swarm can form, particularly focusing on increasing the variety of joints that can be formed and optimizing the topological structure of inter-robot connections. As a result, the snail robot swarm will exhibit greater heterogeneity in the future. The development of advanced swarm control algorithms and strategies for autonomous decision-making are also on the horizon, which could lead to more automated and versatile terrestrial robot swarms. We plan to integrate our previous research on magnet-based localization systems[69] into snail robot swarms. However, direct adaptation poses challenges due to varying magnet layouts in the snail robot's design. Our primary focus is to optimize the number of sensors for accurate localization, ensuring that the computational demands remain within each robot's processing capabilities. By building on the foundation laid by the snail robot swarm, researchers and engineers can continue to advance the field of terrestrial robot swarms and unlock their full potential in unstructured environments.

## Methods

### Morphological study of the land snail

As a key component of our research, we conduct a thorough morphological study on land snails. For this purpose, we procure a selection of White Jade Snails. This study focuses on two key aspects: the general body structure of snails, particularly their natural clustering habits, and the dynamic shape changes of the snail's foot under different external forces. The land snails used in this study comply with the regulations for the Administration of Affairs Concerning Experimental Animals issued by the Institutional Animal Care and Use Committee of Shenzhen.

### Robot fabrication

The spherical shell of the robot is made of thin iron plate with a thickness of 0.8 mm. First, the iron sheet is stamped to form a spherical shell with a diameter of 120 mm. Then, the excess parts are cut off using laser cutting, resulting in the desired shape. The back and sides of the robot feature irregular metal surfaces to ensure smooth transition motions. This integrated irregular surface is fabricated using metal printing technology with mold steel as the material, which also exhibits ferromagnetic properties. The thinnest part of the curved surface has a thickness of 0.8 mm. The metal surface is secured to the robot body using bolt holes. To reduce the overall weight of the robot, the main body components are made from carbon fiber plates that interconnect with one another. Each robot has five motors. Two of them (CHR-GM12-N20 ABHL) are used to drive the caterpillar tracks, one is embedded in the vacuum pump (G2DC1268, maximum vacuum level is −68 Kpa), one (CHR-GM12-N20 ABHL) is used to drive the rotation of the suction cup, and one servo (MG90S, rating torque is 0.2 Nm) controls the vertical movement of the suction cup.

The two magnetic tracks are designed based on two synchronous belts (modulus is 3; the number of teeth is 81). The embedded tension cords within the synchronous belts allow the entire track to withstand higher tension. The manufacturing details for the robot's magnetic tracks can be found in Supplementary Fig. 17. By molding and casting the polymer rubber, the polymer rubber body is prepared, and its grooves are designed to accommodate magnets. The polymer rubber body and the synchronous belts are securely bonded using specialized-rubber adhesive. Finally, silicone rubber gaskets are cut into rectangular shapes slightly larger than the magnet and attached to the magnet's outer surface.

The fabrication process for the sucker with DPS is outlined in Supplementary Fig. 18. By molding and casting polymer silicone with different material ratios, two layers with different hardness are produced. The average thickness of the soft layer is around 1 mm, and the diameter of the stalks on its surface is ~600 μm. A metal connector is placed between the soft and hard layers, and they are bonded together using adhesive. The exposed part of the metal connector will be connected to other parts of the robot body. The metal connector has grooves on its surface to facilitate better adhesion with the silicone sucker.

Each robot has its own microcontroller (STM32f103C8T6) and battery ((lithium polymer battery, 12 V, 500 mAh). Snail robot's electronic and pneumatic architecture can be found in Supplementary Fig. 22. A single-channel pneumatic slip ring (SENRING Electronics Co., Limited, Shenzhen, China) is placed between the air pump and the suction cup to ensure that the suction cup can rotate continuously relative to the robot.

### Performance measurements of the sucker adhesion and the robot

Several key performance metrics can be considered to effectively evaluate the performance of the sucker mechanism and the overall robot system. We measure the maximum normal force, shear force, and torque ($z$ axis) of the sucker using a force gauge (HP-500, HANDPI INSTRUMENTS, Zhejiang, China). More details of sucker performance measurements can be found in Supplementary Fig. 13. For the entire robot, we not only tested the 5 performance metrics of its connectors in both modes but also evaluated the driving capability of the robot, namely the maximum forward or backward driving force and the maximum yaw rotation torque. More details of the connector performance measurement and the driving performance measurement of a snail robot can be found in Supplementary Fig. 14 and Supplementary Fig. 15, respectively.

### Experimental design and data analysis

All individual robot indoor tests are conducted on multiple blue rigid foam boards (500 mm × 500 mm, made of polystyrene). For experiments measuring the maximum gap width the robot can traverse and the maximum step height it can climb, precise measurements of the gap width and step height are taken using a vernier caliper. For the maximum slope angle experiment, a digital angle display is placed on the slope, with the reference baseline being ground. The maximum angle at which the robot can climb without slipping or toppling over is determined as the final result.

In swarm experiments, we assessed the performance of snail robot units in different modes to evaluate their effectiveness in various tasks and environments. All outdoor experiments were conducted in common outdoor scenarios without any special preparation, such as rocky terrain or grassland. All robots are connected to a centralized computer control terminal via Wi-Fi modules. The operator sends commands from the control terminal to control the actions of different robots. After multiple outdoor tests, all robots have not been thoroughly cleaned to demonstrate the durability of their caterpillar mechanisms and suction cup mechanisms against dirt and debris. In multiple experiments, there are instances where one or more robots need to perform self-assembly and self-reconfiguration actions. As we have already demonstrated in the supplementary materials an individual robot meets the requirements for various self-reconfiguration actions, during the experiments, controlling the forward or turning movements of a single robot is sufficient to easily achieve the overall self-reconfiguration of the robot swarm.

All optimization problems mentioned in the papers are solved using MATLAB (MathWorks). Data processing is performed using Python. Unless otherwise specified, uncertainty bounds are provided in the form of means and SD (means ± SD).

## Data availability

All the data required to replicate the results of this study are given in the main article, Supplementary Information and the GitHub repository (https://github.com/Da-Zhao1997/Snail-inspired-robotic-swarms/tree/main/Data).

## Code availability

All the data were processed using custom codes. The code used in this study has been deposited in the public GitHub (https://github.com/Da-Zhao1997/Snail-inspired-robotic-swarms/tree/main/Code) and Zenodo (https://doi.org/10.5281/zenodo.10896716)[70] without any restrictions.

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

## Acknowledgements

We would like to thank Lijun Zong, Guanqi Liang, and Di Wu for their helpful advice during the project. We acknowledge National Natural Science Foundation of China (62073274), Guangdong Basic and Applied Basic Research Foundation (Grant number: 2023B1515020089), and the funding AC01202101103 from the Shenzhen Institute of Artificial Intelligence and Robotics for Society for supporting this research.

## Author contributions

Da Zhao, Haobo Luo, and Tin Lun Lam developed the concept. Da Zhao wrote the manuscript and designed the mechanism, electronics, and control system. Da Zhao, Yuxiao Tu, and Chongxi Meng performed the experiments. Tin Lun Lam guided the overall research program. All authors discussed the results and edited the manuscript.

## Competing interests
The authors declare no competing interests.
