## [Peer Review File · Nature Communications]

REVIEWER COMMENTS

Reviewer #1 (Remarks to the Author):

The Snailbot system, first described in the 2022 ICRA paper “SnailBot: A continuously dockable modular self-reconfigurable robot using rocker-bogie suspension”, was developed as a small mobile robot able to climb atop like robots in freeform formations. The paper presents an interesting iteration upon the robot concept, improving upon the system in meaningful ways. The key innovation presented here is the use of two different attachment systems based on whether a robot is currently moving about a configuration of robots or if it intends to form a secure connection. The work presented here is of very high quality and perhaps its greatest contribution is in demonstrating what such systems are able to do, given current technologies. There is very little to criticize here with regards to the work presented and I recommend that the article be published. However, I have two high-level concerns regarding future extensions of this work, as I describe below. F

My first concern regards the control of the robotic system, specifically the reliance on remote control for their operation. It is not immediately obvious from the work presented how autonomous control could be achieved, especially considering the unique challenges regarding system observability inherent in self-assembling and modular robotic systems. I suspect that the lab’s earlier work on magnet-based localization methods may be applicable here, though the magnetic treads would likely cause many headaches towards their implementation. I know from my own work that integrating sensors into a full-body attachment system can be particularly challenging and requires careful design consideration, so I do not feel that the autonomy issue should be handwaved away as something that can be easily dealt with in the future. I suspect that significant design revisions will be necessary to accommodate full autonomy in the snailbot series of robots and feel that discussion of these challenges and limitations should be added to the paper.

My second general area of concern is the types of structures that snailbots could form. Based on my understanding of the system, each robot could at most connect to two other robots weakly, and only one other robot strongly. This imposes fundamental limitations on the sort of structures that can be built by limiting structure topologies to branching chains (assuming only strong connections). While the authors obviously demonstrate that the sort of structures possible with this system are useful, this nevertheless remains a significant limitation of the system that should be discussed, especially since the “branches” of the structure are limited to only a few units long (I refer here to the cantilever structures shown in Figure 5). I suspect that a key component of the lab’s next work will be integrating the snailbot system into the truss-building system they demonstrated in their “FreeSN” paper.

These critiques should not diminish the overall impressiveness of the work presented, however: these sort of modular and self-assembling robotic systems are very new and remain a relatively unexplored

area of robotics. If someone were to ask for the best example of a modular robotic system, this is likely THE paper that I would point to due to being one of the few examples of such a system operating in unstructured outdoor environments and the many demonstrations of reconfiguration present in the paper.

-Petras Swisler

Reviewer #2 (Remarks to the Author):

This paper demonstrates a novel robotic swarm of terrestrial snail bionics. The main interest of the paper is (i) the authors' implementation of a self-assembling robotic system using the individual robotic hardware platform they developed (albeit with an uncomplicated architecture), and (ii) the impressive demonstration of collaborative behavior between these individuals in the wild achieved through self-assembly.

The strengths of this work are that the most of the paper is fairly well-written and clearly structured, highlighting the experiments presentation (a correct way for this paper); this paper achieves field crawling and self-assembling behaviors that are rarely demonstrated in the relevant literature, which to some extent reflects the effectiveness (robustness) of the proposed design. The downside I think of this work is that the ideas are (at least seemingly) similar to the author's previous paper, so results is not so suprising and the novelty is more or less doubtful. I welcome the author's discussion of the latter point of view in the revised version of the paper.

I think of the paper can be improved in the following ways:

(i) What the paper's aim is to present is a solution for swarm terrestrial robots in the wild; however, the Introduction of the paper did not strike me as well-motivated to consider the snail structure, and although the results of the article illustrate the validity of the structure, I did not find what was presented in the Introduction sufficiently convincing. If snail bionics is not so intuitive, the authors may not need to (conformistically) emphasize its bionic properties, but could honestly articulate other reasons (including more easily achievable mechanism robustness, etc.); if the authors still hold to the bionic viewpoint, I would have liked to see the authors describe the motivation for the use of the snail structure in more a detailed and strong way.

(ii) The paper should be appropriately supplemented with comparisons of this work with previous work by the same authors (e.g., ICRA22) and similar structural self-assemblies previously proposed by the authors, taking into account that their overall ideas are seemed similar. Otherwise, the innovation of contribution of this work at the methodological level is not so clear.

(iii) The authors claim that the design of the connection is important for working in the wild, but this is not reasonably argued to the point where it seems somewhat trivial. I think the authors should have highlighted the functional sections that take center stage for achieving the work's mission (which I think is a good tradition for papers in robotics-related fields), rather than simply flatly stating the obvious.

Overall, I appreciate the systematic work made by the authors (from architecture to algorithm design) as well as the impressive experiments, but the innovativeness of this work compared to the authors' previous work and similar work needs additional discussion, otherwise the reader will not be able to understand where this work fits in the field.

Reviewer #3 (Remarks to the Author):

This paper discusses the design and analysis of reconfigurable robotic swarms, drawing inspiration from the suction mechanism employed by biological snails for attachment. The attachment mechanism includes two different modes depending on the strength of the attachment forces. The authors argue that this innovative design amplifies the swarm's capabilities across a spectrum of challenging indoor and outdoor environments.

The paper is well-structured and from a mechanical design standpoint, it provides detailed insights. Furthermore, the authors provide multiple instances where the swarm successfully accomplishes tasks through inter-swarm attachment. However, there is room for improvement in discussing the experimental aspects, which are somewhat lacking in clarity. An important aspect to consider is that throughout the demonstrations, the authors appear to exert control over individual robots to shape and execute specific structures or behaviors. However, the paper lacks an in-depth discussion regarding the programming methodology employed for these robots or whether any form of inter-robot communication was employed. Addressing these aspects with greater detail and clarity would not only contribute to a more comprehensive understanding of the paper but also amplify its overall impact in the field.

Here are my detailed comments:

1. The authors have another paper (FreeSN ref. 37) which is very similar to what they presented in the current paper with the only difference they have one additional demonstration that shows the capability

of the joined robots to traverse the cobblestone road. Other demonstrations usually show a similar performance which I think does not depend on the environment type (indoor or outdoor). I think the authors should provide more information about the details of the demonstrations. For example, in bridge formation or obstacle climbing, does the new design provide advantages?

One notable point is whether the novel units exclusively enable traversal on rough terrain or if the previous version (similar to what is given in Figure 7E) could also achieve similar results. Clarification in this regard would provide valuable insights into the necessity and efficacy of the joined robot configuration for navigating challenging terrains.

2. I think one of the most pivotal demonstrations illustrating the newfound capabilities of the current swarm configuration is exemplified in Figure 7E. However, the paper appears to lack a comprehensive elucidation of the rationale behind selecting this specific formation, the process that led to the development of the associated gait, the optimal number of robots required for its successful execution, and the primary advantages and functionalities it offers. Please provide more information about these critical aspects.

3. Could you discuss the influence of surface roughness on robot attachment, particularly for those in close proximity to the ground? Additionally, it would be beneficial to elaborate on how variations in surface roughness may impact the distribution of weight within the robot swarm. A more detailed exploration of these factors would contribute significantly to our understanding of the attachment mechanism's performance across diverse terrains.

4. While the authors discussed the strong adhesion mechanism that helps robot-robot attachment, it would be insightful to ascertain whether this adhesion mechanism can also be effectively harnessed to attach the robots to the ground or other surfaces.

5. Please add more references about the biological snail-snail connection mechanism.

6. I think there is room for more in-depth discussion regarding the determination of the optimal number of robots required for specific tasks, taking into consideration factors such as gravitational forces, locomotion efficiency, and obstacle negotiation. Please provide more discussion.

7. I think it is necessary to have further investigation is the impact of surface or environmental cleanliness on the performance of the vacuum suction cups. Understanding how variations in surface conditions, particularly their level of cleanliness, affect the efficacy of the attachment mechanism would offer valuable insights into the real-world applicability and robustness of the technology.

8. Please add more papers about terrestrial swarms, especially Marco Dorigo's papers to references.

Response to reviewers' comments for manuscript

Manuscript No.	: NCOMMS-23-42484
Title	: Snail robots as a new paradigm for swarm robotics in unstructured environments
Authors	: Da Zhao, Haobo Luo, Yuxiao Tu, Chongxi Meng, Tin Lun Lam

Dear Editor,

We would like to express our gratitude for the opportunity to revise our manuscript titled "Snail robots as a new paradigm for swarm robotics in unstructured environments" (Manuscript ID: NCOMMS-23-42484) submitted to Nature Communications. We appreciate the editors' and reviewers' insightful comments and suggestions. We believe that these comments have greatly contributed to enhancing the quality of our paper.

We have undertaken a thorough revision of our manuscript. Every point raised by the reviewers has been addressed carefully, and necessary modifications have been made in the manuscript. Below, please find our responses to each comment made by the reviewers.

Please note that all aspects of the response that are in **green fronts** are sections from reviewers' comments and all that in **blue fonts** are sections that have been added to the revised manuscript.

Response to Reviewer #1

The Snailbot system, first described in the 2022 ICRA paper "SnailBot: A continuously dockable modular self-reconfigurable robot using rocker-bogie suspension", was developed as a small mobile robot able to climb atop like robots in freeform formations. The paper presents an interesting iteration upon the robot concept, improving upon the system in meaningful ways. The key innovation presented here is the use of two different attachment systems based on whether a robot is currently moving about a configuration of robots or if it intends to form a secure connection. The work presented here is of very high quality and perhaps its greatest contribution is in demonstrating what such systems are able to do, given current technologies. There is very little to criticize here with regards to the work presented and I recommend that the article be published. However, I have two high-level concerns regarding future extensions of this work, as I describe below.

Reply: Thank you for your comments. We sincerely appreciate your positive assessment of our Snailbot system and the recognition of its key innovation in dual attachment mechanisms. Your acknowledgment of the system’s versatility and contribution to the field is highly encouraging. We have carefully considered those comments and performed additional studies. We hope the substantial answers can address your concerns. Please find the detailed responses as follows:

Reviewer Point P1.1: My first concern regards the control of the robotic system, specifically the reliance on remote control for their operation. It is not immediately obvious from the work presented how autonomous control could be achieved, especially considering the unique challenges regarding system observability inherent in self-assembling and modular robotic systems. I suspect that the lab’s earlier work on magnet-based localization methods may be applicable here, though the magnetic treads would likely cause many headaches towards their implementation. I know from my own work that integrating sensors into a full-body attachment system can be particularly challenging and requires careful design consideration, so I do not feel that the autonomy issue should be handwaved away as something that can be easily dealt with in the future. I suspect that significant design revisions will be necessary to accommodate full autonomy in the snailbot series of robots and feel that discussion of these challenges and limitations should be added to the paper.

Reply: Thank you for your insightful comments regarding the autonomy and control aspects of our snail robot system. We acknowledge the challenges in achieving autonomous control within self-assembling and modular robotic systems like ours. Currently, our system relies on remote control to demonstrate its fundamental capabilities. However, we acknowledge the significance of autonomy and are investigating potential methods for incorporating it in subsequent versions, including potential applications of our lab’s earlier work on magnet-based localization methods (ref. 69).

For freeform reconfigurable robot swarms, achieving mutual position awareness during connection is a crucial prerequisite for realizing overall automation. Incorporating the magnetic localization algorithm from FreeSN directly into our project does present certain challenges, which indeed warrant a thorough discussion. In FreeSN, the Node module utilizes 42 magnetic sensors, as shown in Fig. R1a. These sensors are somewhat sparsely arranged, but due to the regular arrangement of magnets at the bottom of FreeSN, it achieves effective localization. However, in the case of the snail robot (Fig. R1b), the bottom magnets consist primarily of large central magnets and two side track magnets, leading to an irregular arrangement. Directly using the same number of magnetic sensors as in FreeSN (referenced in ref. 69) may not yield satisfactory results for the snail robot. An alternative could be to increase the number of magnetic sensors to create a denser arrangement. This approach

is exemplified in Fig. R1c, where a very dense sensor layout is employed, capable of generating imaging effects akin to pixels. Moreover, there are constraints related to the onboard computing power of individual robots. If the graph neural network model is too large, it could surpass the available computational capacity. Therefore, the primary challenge in this part of our project is to achieve the highest possible localization accuracy within the limits of available computing power.

We also add some discussions of challenges and limitations in Discussion section:

In Page 32 of the main text, Discussion section: We plan to integrate our previous research on magnet-based localization systems (69) into snail robot swarms. However, direct adaptation poses challenges due to varying magnet layouts in the snail robot’s design. Our primary focus is to optimize the number of sensors for accurate localization, ensuring that the computational demands remain within each robot’s processing capabilities.

Figure R1: **Potential localization method for snail robot based on our lab’s earlier work.** **a** FreeSN’s magnetic localization algorithm. **b** Magnets arrangement on the bottom of the snail robot. **c** An example of densely arranged magnetic sensor array (Reference work link: <https://hackaday.io/project/18518-iteration-8/log/91551-a-third-high-speed-magnetic-imager-tile-draft>).

Reviewer Point P1.2: My second general area of concern is the types of structures that snailbots could form. Based on my understanding of the system, each robot could at most connect to two other robots weakly, and only one other robot strongly. This imposes fundamental limitations on the sort of structures that can be built by limiting structure topologies to branching chains (assuming only strong connections). While the authors obviously demonstrate that the sort of structures possible with this system are useful, this nevertheless remains a significant limitation of the system that should be discussed, especially since the “branches” of the structure are limited to only a few units long (I refer here

to the cantilever structures shown in Figure 5). I suspect that a key component of the lab’s next work will be integrating the snailbot system into the truss-building system they demonstrated in their “FreeSN” paper.

Reply: We appreciate your insightful comments regarding the structural formation capabilities of our Snailbot system. We acknowledge the limitations imposed by the current connection design.

Figure R2: **Two primarily configurations that chain-type freeform modular robots can form. a** Tree-like structure. **b** Graph structure with internal loops.

In our earlier work [1], we referred to the connection type of the chain-type freeform modular robot as a MISO (multiple in-degree, single out-degree) module. While it’s true that this configuration limits the robots to forming primarily serial structures, it also allows for the creation of tree-like or graph structures with internal loops, as shown in Fig. R2. This expands the potential configurations beyond simple linear chains.

We indeed plan to integrate our lab’s other robotic systems, such as FreeSN (ref. 40), with the snail robots. The node module of FreeSN and the outer shell of the snail robot have the same diameter of 12 cm, ensuring their compatibility. Exploring how to combine different types of freeform modular robots, such as FreeSN and snail robots, is an important aspect of our future research.

In the Discussion section of our paper, we also have mentioned future enhancements to snail robots:

In Page 32 of the main text, Discussion section. "In addition, efforts will be made to expand the types of integrated robots that the swarm can form, particularly focusing on increasing the variety of joints that can be formed and optimizing the topological structure of inter-robot connections. As a result, the snail robot swarm will exhibit greater heterogeneity in the future."

Part of this upgrade will involve increasing the types of joints that can be formed upon module connection (currently limited to rotational movements) and enriching the topological

relationships between modules. For example, a future heterogeneous version of the snail robot could strongly connect to two other robots, expanding the current capabilities and structure types.

These critiques should not diminish the overall impressiveness of the work presented, however: these sort of modular and self-assembling robotic systems are very new and remain a relatively unexplored area of robotics. If someone were to ask for the best example of a modular robotic system, this is likely THE paper that I would point to due to being one of the few examples of such a system operating in unstructured outdoor environments and the many demonstrations of reconfiguration present in the paper.

Reply: We would like to express our deep gratitude for your uplifting remarks and for recognizing the significance of our efforts in the realm of modular self-assembling robotic systems. It is a great honor for us to have our paper regarded as a prominent example in this field, and it fuels our commitment to advancing our research even further. We are dedicated to tackling the challenges you highlighted and exploring novel possibilities in modular robotic systems through our ongoing work and future projects.

Response to Reviewer #2

This paper demonstrates a novel robotic swarm of terrestrial snail bionics. The main interest of the paper is (i) the authors' implementation of a self-assembling robotic system using the individual robotic hardware platform they developed (albeit with an uncomplicated architecture), and (ii) the impressive demonstration of collaborative behavior between these individuals in the wild achieved through self-assembly.

The strengths of this work are that the most of the paper is fairly well-written and clearly structured, highlighting the experiments presentation (a correct way for this paper); this paper achieves field crawling and self-assembling behaviors that are rarely demonstrated in the relevant literature, which to some extent reflects the effectiveness (robustness) of the proposed design. The downside I think of this work is that the ideas are (at least seemingly) similar to the author's previous paper, so results is not so suprising and the novelty is more or less doubtful. I welcome the author's discussion of the latter point of view in the revised version of the paper. I think of the paper can be improved in the following ways.

Reply: We appreciate your recognition of the novel aspects and the robustness of the design demonstrated in our paper. We are grateful for your comments on the clarity and structure of the paper, particularly regarding our experiments in field crawling and self-assembling behaviors.

Regarding the concerns about the novelty of our work in relation to our previous publications, we aim to clarify below (Point 2) how this research substantially builds upon and diverges from our earlier work.

In our revised manuscript, we will include a performance comparison chart and a table. These will compare the variety of tasks that can be executed by our robot swarm with other terrestrial modular self-reconfiguring robots. The goal is to clearly delineate the advancements and unique contributions of this work. By doing so, we aim to ensure that the novel aspects of our current research are explicitly highlighted and distinguished from our previous publications.

Reviewer Point P2.1: What the paper's aim is to present is a solution for swarm terrestrial robots in the wild; however, the Introduction of the paper did not strike me as well-motivated to consider the snail structure, and although the results of the article illustrate the validity of the structure, I did not find what was presented in the Introduction sufficiently convincing. If snail bionics is not so intuitive, the authors may not need to (conformistically) emphasize its bionic properties, but could honestly articulate other reasons (including more easily achievable mechanism robustness, etc.); if the authors still hold to the bionic viewpoint, I would have liked to see the authors describe the motivation for the use of the snail structure in more a detailed and strong way.

Reply: Thank you for your valuable feedback on the Introduction of our paper. We appreciate your insights regarding the need for a more detailed and convincing motivation for our choice of snail bionics in the design of our robotic system.

We still hold to the bionic viewpoint. Our robot’s biomimicry primarily encompasses two aspects. The first involves the fundamental structural paradigm, consisting of a spherical shell and propulsion mechanism. The second aspect incorporates a dual-mode connection mechanism, inspired by the primary adhesive method of a snail’s foot under different external forces. We firmly believe that the principles of biomimicry permeate our entire design and should not be overlooked. In response to your comment, we have revised the Introduction to more clearly articulate the rationale behind our design choices.

In Page 4 of the main text, Introduction section:

When developing 3D self-reconfigurable terrestrial robot swarms, the choice of interconnection method among robots holds utmost importance. Freeform, like chain, lattice, truss, and hybrid, stands as a fundamental structural category for modular robot swarms (17, 18). Compared to fixed-position connectors requiring precise dock-to-dock alignment, such as retractable mechanical hooks (20, 27), permanent magnets (10, 22, 25), electromagnets (31), and self-soldering alloys (32), freeform connectors generally offer a much larger acceptance area (33). This broader acceptance proves crucial for large-scale robot swarm deployment, addressing challenges like low sensor precision, manufacturing inaccuracies, and structural deformations (34). In recent years, research has surged around 2D (11, 35-37) and 3D (34, 38-40) freeform robot swarms with high connection success rates. FreeBOT, with its fully spherical shell, is an exceptional freeform robotic system; however, it faces challenges such as a weak single-point connection and limited vertical friction. To address these issues, Zhao and Lam proposed SnailBot (39), a sliding sphere-type robot swarm. Despite improvements, neither FreeBOT nor SnailBot offers sufficient connection strength for tasks like angle-based locomotion or manipulation, which require fixed configurations and considerable shear friction between robots. A potential solution involves a connection mechanism with greater strength, akin to FireAnt3D (34). However, FireAnt3D’s connector has limitations, such as a limited number of cycles and low efficiency. FreeSN (40), the first heterogeneous freeform modular robot, comprises *Node* and *Strut* modules. With a parallel-type structure, FreeSN effectively carries blocks and overcomes obstacles. However, its single module has limited mobility in outdoor environments.

To develop a freeform self-reconfigurable terrestrial robot swarm suitable for outdoor environments, two primary challenges must be addressed. The initial challenge involves

designing a robot with a freeform connector that can effectively operate outdoors. In our pursuit of solutions, we turn to nature for inspiration, leveraging the collective behaviors observed in swarms. Nature’s examples, such as ants forming bridges to cross gaps or gullies (41) and simple components yield high-level behaviors in biological organisms (42, 43), showcase the emergence of remarkable capabilities through collective actions. However, these nature-inspired swarms often lack the mobility required for navigating unstructured environments (11, 34). Recognizing this limitation, we propose exploring land snails. Land snails are gastropod mollusks that possess a unique anatomy (44, 45), allowing them to climb walls, overcome barriers, and navigate uneven surfaces. We develop a novel 3D freeform self-reconfigurable snail robot swarm for field applications, which draws inspiration from the unique anatomical structure of snails. The morphological evolution from a snail to a snail robot is depicted in Fig. 2a. The snail robot’s design incorporates the two primary components of a snail’s body, including the spherical shell and the foot. Furthermore, in nature, snails congregate and even attach to each other for various reasons, such as mating (46), moisture retention, and temperature regulation. The snail robot features the ability to connect to another robot’s ferromagnetic spherical shell using its connection mechanism, which resembles snails’ attachments in nature (Fig. 2b). This expandable capacity enables the formation of larger, more adaptable robotic systems capable of handling a broader array of tasks.

The second challenge involves designing an efficient and stable connector for the snail robot swarm. Traditionally, connectors for self-reconfigurable robots are typically classified as mechanical couplers or magnetic couplers based on the acting forces (47). Here, we classify them based on the level of freedom provided, dividing them into discrete connection (dock-to-dock connection, such as (20, 23)), free connection (connect at any position, but the connection point cannot change once connected, such as (34)), and free transition (connect anywhere and can seamlessly adjust the connection point location, such as (38, 39)). We strive to develop the third type of connector, which boasts the utmost level of freedom. However, this type of connector frequently faces limitations in connection strength. Drawing from nature’s blueprint, the snail robot employs a dual-mode connection mechanism akin to that of a real snail, as illustrated in Fig. 2c. Real snails use mucus adhesion to adhere to substrates (48, 49), enhancing their suction force when they encounter an external pulling force, as demonstrated in experimental studies on snail’s adhesion characteristics (50). Similarly, the snail robot uses magnetic adhesion to connect to other robots’ spherical shells and to transition between robots. When a substantial force is exerted on the robot’s shell, such as when

other snail robots connect to it, the robot extends its vacuum sucker to generate a suction force, to ensure a secure attachment to another robot’s spherical shell. In doing so, we can create a hybrid connector that delivers the highest degree of freedom while maintaining the capability to establish robust connections as required.

Reviewer Point P2.2: The paper should be appropriately supplemented with comparisons of this work with previous work by the same authors (e.g., ICRA22) and similar structural self-assemblies previously proposed by the authors, taking into account that their overall ideas are seemed similar. Otherwise, the innovation of contribution of this work at the methodological level is not so clear.

Figure R3: (the new Supplementary Fig. 7) Comparison with other isomorphic modular self-reconfigurable robot. The ticks of each axis from the graph center to the outward are as follows: Freeform: not free, free connection, free transition; normal force: below 50 N, above 50 N and above 100 N.; tangential force: below 50 N, above 50 N and above 100 N., torsion: below 1 Nm, above 1 Nm and above 5 Nm, dock time: above 5 s, above 1 s and below 1 s., weight: above 1 kg, above 500 g and below 500 g. (Proposed (F+S): F means Free mode and S means Strong mode)

Reply: We appreciate your feedback regarding the need for a clearer comparison with previous works, including our own, to underscore the methodological innovations in this paper. We acknowledge that providing a detailed comparative analysis will help in distinctly highlighting the advancements and unique contributions of our current research.

Table R1: (the new Supplementary Table. 1) Task comparison of 3D modular reconfigurable robots.

Robot	Wild mobility	Self-reconfig.	Self-assembly	Locomotion	Manipulation	Flow	Support
This work	✓	✓	✓	✓	✓	✓	✓
SnailBot (ref. 39)		✓	✓		✓	✓	
Mori (ref. 27)		✓	✓	✓	✓		
FreeSN (ref. 40)		✓	✓	✓	✓		✓
SMORES (ref. 22)		✓	✓	✓			
MTRAN-III (ref. 19)		✓		✓	✓		
FreeBOT (ref. 38)		✓	✓	✓		✓	
FireAnt3D (ref. 34)		✓	✓	✓			✓
RoomBot (ref. 23)		✓		✓	✓		✓
M-Blocks (ref. 25)		✓				✓	
SamBot (ref. 20)		✓		✓	✓		
VTT [2]		✓		✓			✓

¹ Wild mobility means the ability of single robot’s mobility in the wild.

² All capabilities of the robot are based on what appears in their related papers.

In response to your concerns about the novelty and methodological advancements of our current work compared to our previous studies, particularly those presented in ICRA 2022, we have made the following clarifications and additions:

The most significant contribution of this work lies in the development of a ground-based robot swarm designed for outdoor use. This marks a substantial step forward from our prior research. SnailBot (ICRA2022) employs an actively lifting six-wheel rocker-bogie mechanism to execute self-reconfiguration actions. However, the rocking-arm mechanism demonstrates superior obstacle-surmounting capabilities only when utilizing passive joints. Moreover, the mecanum wheels used in SnailBot are typically considered not suitable for wild unstructured environments. We replace this actively lifting mechanism used in our earlier work with a more robust and efficient tracked chassis. This new design simplifies the structure and enhances the robot’s ability to navigate obstacles. Another key innovation in this work is the implementation of a dual-mode connection mechanism. This mechanism, inspired by observations of real snails, allows for different connection modes based on task requirements, greatly enhancing the versatility and functionality of the robot swarm. The enhancements

introduced in this work not only improve individual robot performance but also expand the complexity and variety of tasks that can be performed collaboratively by robot swarms. This represents a significant breakthrough compared to our previous capabilities.

To clearly illustrate these advancements, we have included a performance comparison chart in the manuscript (Fig. R3). This chart benchmarks six key indicators against several existing homogenous freeform modular self-reconfiguring robots. Additionally, we have added a table comparing the variety of tasks that can be executed by our robot swarm against other terrestrial modular self-reconfiguring robots (Table. R1). This comparison highlights the superior capabilities of our current work.

We believe these additions and clarifications will adequately address your concerns regarding the novelty and methodological contributions of our work. We add this performance comparison chart and task comparison table to the Supplementary file, providing a clear distinction from our previous publications (see Supplementary Note 5).

Reviewer Point P2.3: The authors claim that the design of the connection is important for working in the wild, but this is not reasonably argued to the point where it seems somewhat trivial. I think the authors should have highlighted the functional sections that take center stage for achieving the work’s mission (which I think is a good tradition for papers in robotics-related fields), rather than simply flatly stating the obvious.

Reply: We believe that our paper does not explicitly state that the design of the connection is crucial for working in the wild. Rather, our emphasis lies in elucidating the importance of designing a reconfigurable and collaborative robot swarm tailored for outdoor use. The consideration of connection design is rooted in the overarching goal of creating a robot swarm capable of effective deployment in the wild.

However, we also acknowledge your point regarding ‘I think the authors should have highlighted the functional sections that take center stage for achieving the work’s mission.’ In our revised manuscript, we provide a detailed exposition of the capabilities required for individual robots and the swarm as a whole. We emphasize and elucidate why these capabilities are crucial for the success of our mission.

In Page 3 of the main text, Introduction section:

In the context of large-scale deployment in outdoor environments, reconfigurable terrestrial robot swarms exhibit significant potential for operation in unstructured settings. These adaptable robots, when functioning as individual units, demonstrate the capability to explore and maneuver in diverse outdoor scenarios. The emphasis on the single robot’s field mobility ensures the overall system’s flexibility and agility. Furthermore, the incorporation of a robust connection mechanism becomes pivotal, ensuring that

when the robot swarm assembles into a cohesive unit, it attains heightened robustness. This multifaceted approach highlights the synergy between individual mobility and robust interconnectivity, crucial for the successful deployment and operation of reconfigurable terrestrial robot swarms in dynamic outdoor environments (Fig. 1).

Overall, I appreciate the systematic work made by the authors (from architecture to algorithm design) as well as the impressive experiments, but the innovativeness of this work compared to the authors' previous work and similar work needs additional discussion, otherwise the reader will not be able to understand where this work fits in the field.

Reply: We sincerely thank you for recognizing the systematic nature of our work and the efforts put into our experiments. We appreciate your feedback highlighting the need for a more explicit discussion on the novelty and innovation of our current research, particularly in relation to our previous work and similar studies in the field.

In response to your valuable input, we plan to revise the manuscript to include a detailed comparison and discussion that clearly delineates how this work advances beyond our previous research. We aim to articulate the unique contributions of our current study and its significance within the broader context of robotic systems research. This will include highlighting the specific methodological and technological advancements we have achieved and how they contribute to the field.

We are committed to ensuring that the revised version of the paper provides a clear understanding of where our work fits in the field and its innovative aspects. We believe this enhancement will significantly improve the paper's clarity and help readers appreciate the unique value of our research.

Response to Reviewer #3

This paper discusses the design and analysis of reconfigurable robotic swarms, drawing inspiration from the suction mechanism employed by biological snails for attachment. The attachment mechanism includes two different modes depending on the strength of the attachment forces. The authors argue that this innovative design amplifies the swarm’s capabilities across a spectrum of challenging indoor and outdoor environments.

The paper is well-structured and from a mechanical design standpoint, it provides detailed insights. Furthermore, the authors provide multiple instances where the swarm successfully accomplishes tasks through inter-swarm attachment. However, there is room for improvement in discussing the experimental aspects, which are somewhat lacking in clarity. An important aspect to consider is that throughout the demonstrations, the authors appear to exert control over individual robots to shape and execute specific structures or behaviors. However, the paper lacks an in-depth discussion regarding the programming methodology employed for these robots or whether any form of inter-robot communication was employed. Addressing these aspects with greater detail and clarity would not only contribute to a more comprehensive understanding of the paper but also amplify its overall impact in the field. Here are my detailed comments:

Reply: We appreciate your positive feedback on the structure and mechanical design insights of our paper, and thank you for your constructive comments regarding the experimental aspects.

We agree that including these details will contribute significantly to a more comprehensive understanding of our work. Enhancing the clarity and depth of discussion in these areas will also amplify the overall impact of the paper in the field of reconfigurable robotic swarms. We are committed to making these improvements in the revised version of the manuscript.

Reviewer Point P3.1: The authors have another paper (FreeSN ref. 40) which is very similar to what they presented in the current paper with the only difference they have one additional demonstration that shows the capability of the joined robots to traverse the cobblestone road. Other demonstrations usually show a similar performance which I think does not depend on the environment type (indoor or outdoor). I think the authors should provide more information about the details of the demonstrations. For example, in bridge formation or obstacle climbing, does the new design provide advantages? One notable point is whether the novel units exclusively enable traversal on rough terrain or if the previous version (similar to what is given in Figure 8E) could also achieve similar results. Clarification in this regard would provide valuable insights into the necessity and efficacy of the joined robot configuration for navigating challenging terrains.

Reply: We thank you for your comments and for pointing out the need for a clearer

Figure R4: **(the new Supplementary Fig. 7) Comparison with other isomorphic modular self-reconfigurable robot.** The ticks of each axis from the graph center to the outward are as follows: Freeform: not free, free transition, free connection; normal force: below 50 N, above 50 N and above 100 N.; tangential force: below 50 N, above 50 N and above 100 N., torsion: below 1 Nm, above 1 Nm and above 5 Nm, dock time: above 5 s, above 1 s and below 1 s., weight: above 1 kg, above 500 g and below 500 g. (Proposed (F+S): F means Free mode and S means Strong mode)

distinction between our current work and our previous paper (FreeSN ref. 40). While there are certain similarities between the two papers, we agree that a more detailed explanation of the advancements and unique aspects of the new design is necessary.

We believe that the main distinctions and improvements of the presented snail robot swarm compared to FreeSN can be summarized in two key aspects. Firstly, the individual mobility of a single snail robot allows it to navigate various terrains in the wild. This enhanced maneuverability is particularly valuable in scenarios where a single robot needs to traverse narrow passages or demonstrate unique movements, as showcased in Figures 1 and 7. Such capabilities are not inherent in other reconfigurable ground robot swarms, including FreeSN. Additionally, while both snail robot and FreeSN can navigate through some challenging terrains, the individual mobility of snail robots enables specific modules to detach and perform tasks independently, as illustrated in Figure 8g — a capability not achievable by FreeSN due to limitations in individual module mobility. Moreover, the snail

Table R2: (the new Supplementary Table. 1) Task comparison of 3D modular reconfigurable robots.

Robot	Wild mobility	Self-reconfig.	Self-assembly	Locomotion	Manipulation	Flow	Support
This work	✓	✓	✓	✓	✓	✓	✓
SnailBot (ref. 39)		✓	✓		✓	✓	
Mori (ref. 27)		✓	✓	✓	✓		
FreeSN (ref. 40)		✓	✓	✓	✓		✓
SMORES (ref. 22)		✓	✓	✓			
MTRAN-III (ref. 19)		✓		✓	✓		
FreeBOT (ref. 38)		✓	✓	✓		✓	
FireAnt3D (ref. 34)		✓	✓	✓			✓
RoomBot (ref. 23)		✓		✓	✓		✓
M-Blocks (ref. 25)		✓				✓	
SamBot (ref. 20)		✓		✓	✓		
VTT [2]		✓		✓			✓

¹ Wild mobility means the ability of single robot’s mobility in the wild.

² All capabilities of the robot are based on what appears in their related papers.

robot swarm can execute flow motions, rapidly altering the inter-module connections, a feat beyond the capabilities of individual FreeSN modules, which would require at least three modules to achieve.

Secondly, the augmented connection robustness of the snail robot swarm is crucial for reconfigurable modular robot swarms. The ability demonstrated in Figure 8E is primarily attributed to the increased connection strength and torque. Theoretically, under the same module count, the snail robot swarm can traverse wider gaps compared to FreeSN (Figure 8g). These improvements underscore the necessity and efficacy of the joined robot configuration for navigating challenging terrains, offering advantages in terms of enhanced individual mobility, independent task execution, and improved connection robustness.

To clearly illustrate the advancements in current work, we have also included a performance comparison chart in the manuscript (Fig. R4). This chart benchmarks six key indicators against several existing homogenous freeform modular self-reconfiguring robots. Additionally, we have added a table comparing the variety of tasks that can be executed by our robot swarm against other terrestrial modular self-reconfiguring robots (Table. R2).

This comparison highlights the superior capabilities of our current work.

We believe these additions and clarifications will adequately address your concerns regarding the novelty and methodological contributions of our work. We add this performance comparison chart and task comparison table to the Supplementary file, providing a clear distinction from our previous publications (see Supplementary Note 5).

Reviewer Point P3.2: I think one of the most pivotal demonstrations illustrating the newfound capabilities of the current swarm configuration is exemplified in Figure 7E. However, the paper appears to lack a comprehensive elucidation of the rationale behind selecting this specific formation, the process that led to the development of the associated gait, the optimal number of robots required for its successful execution, and the primary advantages and functionalities it offers. Please provide more information about these critical aspects.

Reply: We appreciate your suggestions in more discussions on determining the optimal number of robots required for a specific task.

In the face of specific tasks, how MSRR determines the most appropriate number of modules and the connection relationship between modules is a branch of the research field of automatic robot design that has long been concerned but is not well solved. Specific tasks are often modeled by researchers in this field as simulation environments that take into account different task factors such as gravitational forces, locomotion efficiency and obstacle negotiation. These simulation environments serve as platforms to evaluate the performance scores of MSRRs with different module numbers or configurations, employing uniform control methods such as reinforcement learning (ref. 63) or MPC (Model Predictive Control) (ref. 64). By evaluating these configurations, a guidance is provided to optimize module numbers and arrangements.

Earlier optimization methods have employed genetic algorithms (65, 66), library addressing (ref. 67), gradient descent (ref. 68), among others. Recent studies have started leveraging neural networks for heuristic searches, such as RoboGrammar (ref. 64) and Transform2Act (ref. 63). Nonetheless, current research faces challenges due to the time-intensive nature of configuration evaluations and the discrete nature of optimization parameters. These hurdles are the focal points our research group aims to tackle in our upcoming work.

Our initial discourse on this research topic is presented as above, and we have added a discussion in the main text as well.

In Page 31 of the main text, Discussion section: In determining the optimal number of modules and their interconnection for specific tasks, a key challenge in automatic robot design, researchers model tasks in simulation environments. These environments consider factors like gravitational forces and obstacle negotiation, and various module numbers or configurations are evaluated using methods like reinforcement learning

(63) or Model Predictive Control (MPC) (64). Optimization methods have historically included genetic algorithms (65, 66), library addressing (67), and gradient descent (68). Recent studies leverage neural networks, such as RoboGrammar (64) and Transform2Act (63). Challenges persist due to time-intensive evaluations and discrete optimization parameters, areas our research group aims to address in future work.

Reviewer Point P3.3: Could you discuss the influence of surface roughness on robot attachment, particularly for those in close proximity to the ground? Additionally, it would be beneficial to elaborate on how variations in surface roughness may impact the distribution of weight within the robot swarm. A more detailed exploration of these factors would contribute significantly to our understanding of the attachment mechanism's performance across diverse terrains.

Reply: We thank you for bringing attention to the significant aspect of surface roughness and its impact on robot attachment and weight distribution within the swarm. We recognize the criticality of these factors in understanding the performance of our attachment mechanism across diverse terrains.

This question pertains to the issue of dynamic gravity stability. If the ground is relatively rough, the robot's center of gravity may shift forward, resulting in a greater torque at the front. On the other hand, if the ground is less rough, the robot may sway around the geometric center. The article "Distributed Prediction of Unsafe Reconfiguration Scenarios of Modular Robotic Programmable Matter" (ref. 62) addresses static gravity stability, and it is relevant to cite it. Extending from this, it is worth noting that dynamic gravity stability remains an unsolved challenge. From an intuitive perspective gained through experimental work, the Modular Self-Reconfigurable Robot (MSRR) adopts configurations close to the ground, such as snake-shaped or wheeled tripod configurations. These configurations may involve more ground contact, necessitating stable connection points capable of withstanding collision forces.

As the roughness of the terrain increases, the stability requirements for the connections between modules become higher. This is because locomotion on rough terrain may demand higher rotational torque. Without suction cups to enhance stability, modules may experience instability when facing terrains like cobblestones, where the connections between modules may not withstand greater driving torque and could break. In general, on rough terrain, MSRR favors configurations where the torque is biased towards the geometric center. For instance, the tripod configuration demonstrated in our demo showcases effective movement on cobblestones. This configuration allows the front two arms to exert greater torque to overcome ground friction. It's crucial to highlight that this discussion falls within the realm of dynamic gravity stability research, and while we provide some insights, it remains an area

for future exploration in our work.

We also add a discussion on this issue in main text.

In Page 27 of the main text, Traverse the Cobblestone Road subsection: *As the roughness of the terrain increases, the stability requirements for the connections between modules become higher. This is because locomotion on rough terrain may demand higher rotational torque. Without suction cups to enhance stability, modules may experience instability when facing terrains like cobblestones, where the connections between modules may not withstand greater driving torque and could break. In general, on rough terrain, MSRR favors configurations where the torque is biased towards the geometric center. For instance, the tripod configuration demonstrated in our demo showcases effective movement on cobblestones. This configuration allows the front two arms to exert greater torque to overcome ground friction.*

Reviewer Point P3.4: *While the authors discussed the strong adhesion mechanism that helps robot-robot attachment, it would be insightful to ascertain whether this adhesion mechanism can also be effectively harnessed to attach the robots to the ground or other surfaces.*

Reply: Thank you for your insightful suggestion regarding the potential application of our strong adhesion mechanism beyond robot-robot attachment. We have carefully considered the feasibility of utilizing this mechanism for attaching robots to the ground or other surfaces, and we appreciate the opportunity to provide further clarification.

The working mechanism of snail robot’s two modes is illustrated in Fig. R5a. In free mode, the suction cup is elevated, and upon switching to strong mode, the entire connection mechanism extends through gears, bringing the suction cup into contact with the spherical shell. Fig. R5b illustrates the presence of position-constraining mechanisms, ensuring minimal impact on the lifting gear mechanism even under substantial external tensile force. This design advantage results in a simple and lightweight lifting system. However, a potential limitation is that the suction cup cannot extend to the same horizontal plane as the magnetic tracks. Therefore, when the robot operates on a flat surface, the strong connection mechanism cannot adhere to the surface.

We did explore the possibility of extending the suction cup further to enhance adhesion on flat surfaces, as depicted in Fig. R5c. This would provide increased stability and reduce the risk of tipping. However, after careful consideration, we decided against this modification. Given that our snail robot group primarily operates outdoors in varied and complex environments, achieving consistent and tight adhesion to overly complex surfaces

Figure R5: **Discussion of the strong adhesion mechanism adhering to other surfaces.** **a** Working mechanism of two modes. **b** The position-limiting mechanism inside the strong adhesion mechanism. **c** Strong adhesion to the ground can enhance stability and reduce the risk of tipping in some cases. **d** A snail robot adhere to iron street light pole.

proved challenging. Therefore, our focus remained on enabling robots to establish a strong connection with the spherical shell of another robot.

Nevertheless, it is important to note that, in specific scenarios, such as encounters with iron lampposts, the snail robots demonstrate the capability to adhere and climb, as illustrated in Fig. R5d.

We hope this explanation clarifies our considerations and choices in designing the snail robot's adhesion mechanism.

We also add a discussion on this issue in main text.

In Page 11 of the main text, Mechanism Overview section: We did explore the possibility of extending the suction cup further to enhance adhesion on flat surfaces. This would provide increased stability and reduce the risk of tipping. However, after careful

consideration, we decided against this modification. Given that our snail robot group primarily operates outdoors in varied and complex environments, achieving consistent and tight adhesion to overly complex surfaces proved challenging.

Reviewer Point P3.5: Please add more references about the biological snail-snail connection mechanism.

Reply: Thank you for your comments. While there are several studies on snails, specifically in relation to their interaction with environments, I couldn't find any research papers specifically focused on "snail-to-snail connection mechanisms" in the context of direct interactions or communication between individual snails. However, I do find research related to biological snail connection mechanism with the environment. I have add two references about it to the manuscript.

Reference [48]: J. Li, X. Peng, C. Ma, Z. Song, J. Liu, Response mechanisms of snails to the pulling force and its potential application in vacuum suction, *Journal of the Mechanical Behavior of Biomedical Materials* 124, 104840 (2021).

Reference [49]: N. J. Shirtcliffe, G. McHale, M. I. Newton, Wet adhesion and adhesive locomotion of snails on anti-adhesive non-wetting surfaces, *PLoS One* 7, e36983 (2012).

In Page 7 of the main text, Introduction section: Real snails use mucus adhesion to adhere to substrates (48,49), enhancing their suction force when they encounter an external pulling force, as demonstrated in experimental studies on snail's adhesion characteristics (50).

Reviewer Point P3.6: I think there is room for more in-depth discussion regarding the determination of the optimal number of robots required for specific tasks, taking into consideration factors such as gravitational forces, locomotion efficiency, and obstacle negotiation. Please provide more discussion.

Reply:

We have identified that this question bears similarity to **Reviewer Point 3.2**, and as such, we show a similar answer here:

We appreciate your suggestions in more discussions on determining the optimal number of robots required for a specific task.

In the face of specific tasks, how MSRR determines the most appropriate number of modules and the connection relationship between modules is a branch of the research field of automatic robot design that has long been concerned but is not well solved. Specific tasks are often modeled by researchers in this field as simulation environments that take into account different task factors such as gravitational forces, locomotion efficiency and obstacle

negotiation. These simulation environments serve as platforms to evaluate the performance scores of MSRRs with different module numbers or configurations, employing uniform control methods such as reinforcement learning (ref. 63) or MPC (Model Predictive Control) (ref. 64). By evaluating these configurations, a guidance is provided to optimize module numbers and arrangements.

Earlier optimization methods have employed genetic algorithms (65, 66), library addressing (ref. 67), gradient descent (ref. 68), among others. Recent studies have started leveraging neural networks for heuristic searches, such as RoboGrammar (ref. 64) and Transform2Act (ref. 63). Nonetheless, current research faces challenges due to the time-intensive nature of configuration evaluations and the discrete nature of optimization parameters. These hurdles are the focal points our research group aims to tackle in our upcoming work.

Our initial discourse on this research topic is presented as above, and we have added a discussion in the main text as well.

In Page 31 of the main text, Discussion section: In determining the optimal number of modules and their interconnection for specific tasks, a key challenge in automatic robot design, researchers model tasks in simulation environments. These environments consider factors like gravitational forces and obstacle negotiation, and various module numbers or configurations are evaluated using methods like reinforcement learning (63) or Model Predictive Control (MPC) (64). Optimization methods have historically included genetic algorithms (65, 66), library addressing (67), and gradient descent (68). Recent studies leverage neural networks, such as RoboGrammar (64) and Transform2Act (63). Challenges persist due to time-intensive evaluations and discrete optimization parameters, areas our research group aims to address in future work.

Reviewer Point P3.7: I think it is necessary to have further investigation is the impact of surface or environmental cleanliness on the performance of the vacuum suction cups. Understanding how variations in surface conditions, particularly their level of cleanliness, affect the efficacy of the attachment mechanism would offer valuable insights into the real-world applicability and robustness of the technology.

Reply: Thank you for your valuable suggestion to investigate the impact of surface and environmental cleanliness on the performance of our vacuum suction cups. We have conducted additional studies, and the results are now incorporated into our revised manuscript.

In our experiments, we focused on three typical outdoor contaminants: weeds, dry soil or dust, and dirty water, as illustrated in Fig. R6a-c. To establish baseline vacuum levels, we initially tested the suction cups in a contaminant-free environment (Fig. R6d) and subsequently introduced each contaminant separately to observe their effects.

With a substantial presence of weeds between the suction cup and the shell surface, the vacuum level drastically reduced to nearly zero, indicating a loss of functionality (Fig. R6e). However, when only a minimal amount of weeds was present, the vacuum level was approximately half of the clean condition (Fig. R6f). When testing with dust, we observed a similar pattern. A large quantity of dust completely negated the suction force (Fig. R6g), while a lesser amount reduced the vacuum level to about half (Fig. R6h). In contrast, the introduction of dirty water sprinkled on the shell surface had a negligible impact on the suction cups' performance (Fig. R6i).

Figure R6: **Impact of Surface Cleanliness on Vacuum Suction Performance.** **a** Weeds impurities. **b** Dry soil and dust impurities. **c** Dirty water impurities. **d** Vacuum suction performance with no impurities (clean surface). **e** Vacuum suction performance with high weed contamination. **f** Vacuum suction performance with low weed contamination. **g** Vacuum suction performance with high dust contamination. **h** Vacuum suction performance with low dust contamination. **i** Vacuum suction performance with dirty water contamination.

Our findings indicate that while the presence of a significant amount of weeds or dust between the suction cup and the shell surface can reduce the vacuum level, it's important

to note that such extreme conditions are not commonly encountered in typical operational environments. In scenarios with a minimal amount of weeds or dust, which are more representative of real-world conditions, the impact on the suction cups was limited, reducing the vacuum level to about half of the optimal performance. Interestingly, the presence of dirty water had a negligible effect on the performance, demonstrating a degree of resilience of the suction cups in wet conditions.

Figure R7: (the updated Fig. 6) Properties of snail robot in strong mode. **a** Analysis of DPS's reactive force against the normal force. **b** Analysis of DPS's reactive force against the shear force. **c** Analysis of DPS's reactive force against the torque along the z axis. **d** Hardness parameters of the multi-layer sucker and size parameters of the DPS. **e** Comparison of connection strength and driving capability in free mode and strong mode. (Except for Torque and Turn, the units for other data are in N.) **f** The maximum number of connectable modules for the two cantilever structures.

These findings indicate that while surface cleanliness does impact the performance of the

vacuum suction cups, the effect is limited in scope. In most practical scenarios, where extreme contamination is nearly unlikely, the suction cups maintain a significant level of functionality. This demonstrates that while there is room for improvement, the current design of the vacuum suction cups is reasonably robust for typical operational conditions. Nevertheless, in case of working in extreme conditions, we also contemplate potential enhancements. For instance, a future design iteration could incorporate a reverse-jet functionality in the suction cups. This feature would clear impurities from the spherical shell before engaging in a strong connection, ensuring optimal performance even in challenging environments.

We have incorporated these experiments into the main text, contributing to a more comprehensive presentation of the performance of the connection mechanism:

In Page 22 of the main text, Strong Mode Connection and Motion subsection: The investigation into the impact of surface and environmental cleanliness on the performance of the strong connection mechanism is imperative. Our experiments (Figures 6g-h) revealed that the presence of significant contaminants such as weeds or dust between the suction cup and the shell surface substantially reduced the vacuum level, leading to a loss of functionality. However, under more realistic conditions with minimal contaminants, the suction cups demonstrated resilience, maintaining approximately half of the optimal performance. Notably, the introduction of dirty water had negligible effects, indicating a degree of robustness in wet conditions. These findings underscore the importance of considering real-world environmental factors in evaluating the functionality of the vacuum suction cups, with the current design proving reasonably robust for typical operational conditions. Nevertheless, in case of working in extreme conditions, we also contemplate potential enhancements. For instance, a future design iteration could incorporate a reverse-jet functionality in the suction cup. This feature would clear impurities from the spherical shell before engaging in a strong connection, ensuring optimal performance even in challenging environments.

Reviewer Point P3.8: Please add more papers about terrestrial swarms, especially Marco Dorigo's papers to references.

Reply: Thank you for your comments. I have add three more papers about robot swarms to references.

Reference [8]: E. Bonabeau, M. Dorigo, G. Theraulaz, Swarm intelligence: from natural to artificial systems, no. 1 (Oxford university press, 1999).

Reference [14]: M. Dorigo, D. Floreano, L. M. Gambardella, F. Mondada, S. Nolfi, T. Baaboura, M. Birattari, M. Bonani, M. Brambilla, A. Brutschy, et al., Swarmanoid: a

novel concept for the study of heterogeneous robotic swarms, *IEEE Robotics & Automation Magazine* 20, 60–71 (2013)

Reference [15]: M. Brambilla, E. Ferrante, M. Birattari, M. Dorigo, Swarm robotics: a review from the swarm engineering perspective, *Swarm Intelligence* 7, 1–41 (2013).

In Page 2 of the main text, Introduction section: This remarkable feature of swarm intelligence allows these groups to achieve greater efficiency, robustness, and adaptability, ultimately enhancing their chances of survival and success in their respective environments (8).

In Page 3 of the main text, Introduction section: While numerous indoor terrestrial robot swarms have been developed for operation on flat surfaces (11-15), they are often ill-suited for unstructured environments with steps, ditches, and varying surface materials.

References

- [1] H. Luo, T. L. Lam, Auto-optimizing connection planning method for chain-type modular self-reconfiguration robots, *IEEE Transactions on Robotics* **39**, 1353–1372 (2022).
- [2] E. Park, J. Bae, S. Park, J. Kim, M. Yim, T. Seo, Reconfiguration solution of a variable topology truss: Design and experiment, *IEEE Robotics and Automation Letters* **5**, 1939–1945 (2020).

REVIEWERS' COMMENTS

Reviewer #1 (Remarks to the Author):

I feel the changes sufficiently address my concerns and I have no issue with the paper being submitted as-is.

Reviewer #2 (Remarks to the Author):

The authors made significant efforts to address the comments. All of the comments have been addressed. I support that this paper should be considered for publication at the discretion of the editor.

Reviewer #3 (Remarks to the Author):

Thanks for resolving most of the comments. One of my comments is not answered clearly (Point 3.2). In that comment, I understand there are challenges in that domain, resolving them may be out of the scope of this paper. But at least in the text, you can discuss how you chose this configuration (even if it is empirical) of the swarm which creates a different robot (a legged robot with a body) and locomote together. When I watched the SI movie 5, it seemed robots react to changes in the configuration and correct their rotation speed. There is something here, but I couldn't find the discussion about this demonstration in the paper. As I said in my review, this is the most pivotal demonstration illustrating the newfound capabilities of the current swarm configuration.

Also, can you please ensure that all Supplementary Information movies are appropriately cited in the corresponding figures within the paper?

Final revision response to reviewers' comments for manuscript

Manuscript No. : NCOMMS-23-42484
Title : Snail-inspired robotic swarms: a hybrid connector drives collective adaptation in unstructured outdoor environments
Authors : Da Zhao, Haobo Luo, Yuxiao Tu, Chongxi Meng, Tin Lun Lam

Dear Editor,

Thank you for considering our manuscript titled "Snail-inspired robotic swarms: a hybrid connector drives collective adaptation in unstructured outdoor environments" for publication in Nature Communications. We sincerely appreciate the opportunity to address the reviewers' comments and concerns in this final revision.

In particular, we have focused on providing a more detailed response to point 3.2 of reviewer #3's report regarding the choice of the specific configuration employed in our study. Furthermore, we select an alternative title that does not incorporate the expression "new paradigm".

Please note that all aspects of the response that are in **green fronts** are sections from reviewers' comments and all that in **blue fonts** are sections that have been added to the revised manuscript.

Response to Reviewer #1

I feel the changes sufficiently address my concerns and I have no issue with the paper being submitted as-is.

Reply: Thank you for your support and positive evaluation of our work.

Response to Reviewer #2

The authors made significant efforts to address the comments. All of the comments have been addressed. I support that this paper should be considered for publication at the discretion of the editor.

Reply: Thank you for your support and positive evaluation of our work.

Response to Reviewer #3

Reviewer Point P3.1: Thanks for resolving most of the comments. One of my comments is not answered clearly (Point 3.2). In that comment, I understand there are challenges in that domain, resolving them may be out of the scope of this paper. But at least in the text, you can discuss how you chose this configuration (even if it is empirical) of the swarm which creates a different robot (a legged robot with a body) and locomote together. When I watched the SI movie 5, it seemed robots react to changes in the configuration and correct their rotation speed. There is something here, but I couldn't find the discussion about this demonstration in the paper. As I said in my review, this is the most pivotal demonstration illustrating the newfound capabilities of the current swarm configuration.

Reply: Thank you very much for your advice! I'm sorry that my last reply did not fully answer your question. I hope that the second reply below can answer your concerns.

Figure R1: How can we choose the configuration of the swarm that creates a different robot and locomote together. We use reinforcement learning to control each candidate configuration to perform tasks such as locomotion in the simulation environment and use the final reward as the performance score of the candidate configuration. (a)(b)(c) A certain configuration of MSRR is stuck in the gap terrain. Through the configuration optimization method we studied, a new connection relationship is output, so that the gap terrain can be crossed. (d)(e)(f) Similarly, a serpentine-shaped MSRR is blocked by a narrow channel. Through the configuration optimization method we studied, the connection position of each module can be adjusted. Thereby the entire configuration is compressed to drill through the narrow channel.

We choose the configuration of the swarm based on the final reward of many candidate configurations controlled by reinforcement learning in the simulation environment. Taking

the execution of the locomotion task in the gap terrain as an example, the final reward of the configuration with the connection relationship shown in Fig. R1(a) is significantly smaller than the final reward of the configuration with the connection relationship shown in Fig. R1(c). We selected the configuration in the SI movie 5 after evaluating a large number of candidate configurations in a simulated flatland environment. The rationale behind selecting this specific formation or configuration is that the performance score gap between the configurations in the simulation environment can roughly evaluate the actual performance score gap of the real robots. For example, although the two configurations shown in Fig. R1(d) and Fig. R1(f) use simplified simulation models that only simulate the size and joint positions of physical modules, the physical modular robot using the configuration shown in Fig. R1(f) will perform better on real locomotion tasks, which is consistent with the conclusion obtained from the simulation.

Our group has a research focus on configuration selection and design. I was inspired by the preliminary results of this research when I was conducting experiments on the SI movie 5. This research proposes a configuration design framework for MSRR, including connectivity design and junction design, which are used to optimize discrete parameters and continuous parameters in the configuration, such as connection relationships and connection positions, as shown in Fig. R1(b) and Fig. R1(e) respectively. For example, the configuration optimized in simulation shown in Fig. R1(c) can better locomotion on gap terrain or flat terrain, which is exactly the configuration we used in Movie 5. This research will be made public soon and serve as a formal process that led to the development of the associated gait such as the one in the SI movie 5. The optimal number of robots required for its successful execution is also determined in the process described above by comparing a large number of candidate configurations using reinforcement learning.

The primary advantages and functionalities our chosen configuration offers are: (1) The two legs on both sides of this configuration use the shell of the snail robot as wheels, which can ensure that the robot rotates at a differential speed and avoids obstacles. (2) The two legs of this configuration extend a lot, ensuring that the entire large robot walks stably and will not roll over. (3) The end of the tail module is the crawler surface of a robot module, which can also provide power for the movement of the entire large robot. These advantages are our empirical summary from simulation results and physical experiments, and can also be used as a reference for future empirical configuration selection. As you said in the comment the robots in the SI movie 5 can react to changes in the configuration and correct their rotation speed. This is also because the configuration is more stable as mentioned above, and in the simulation we also found that reinforcement learning can control this configuration to react and correct more stably.

Based on the above analysis and your suggestions, we updated the discussion as below.

In Page 31 of the main text, Discussion section: In determining the optimal number of modules and their interconnection for specific tasks, we choose the configuration of the swarm based on the final reward of many candidate configurations controlled by reinforcement learning in the simulation environment. For example, the configuration in Fig. 8e is selected after evaluating a large number of candidate configurations in a simulated flatland environment. The rationale behind selecting this specific configuration is that the performance score gap between the configurations in the simulation environment can roughly evaluate the actual performance score gap of the real robots. From the empirical analysis of physical experiments, this configuration does have multiple advantages. For example, the two legs on both sides of this configuration can rotate at differential speeds to avoid obstacles, while ensuring that the entire large robot walks stably. Methods for configuration selection and optimization similar to those described above have historically included genetic algorithms (63, 64), library addressing (65), and gradient descent (66). Recent studies leverage neural networks, such as RoboGrammar (67) and Transform2Act (68). Challenges persist due to time-intensive evaluations and discrete optimization parameters, areas our research group aims to address in future work.

Reviewer Point P3.2: Also, can you please ensure that all Supplementary Information movies are appropriately cited in the corresponding figures within the paper?

Reply: Thank you for your comments. I believe it is not necessary to cite the Supplementary Information movies in the corresponding figures within the paper, as the link for all the final videos is provided in figure 1 in the main text. Additionally, the Supplementary Information file already includes the YouTube links to the videos at the beginning.